# Extinction risk and conservation of the world's sharks and rays

Nicholas K Dulvy[1,2]*, Sarah L Fowler[3], John A Musick[4], Rachel D Cavanagh[5], Peter M Kyne[6], Lucy R Harrison[1,2], John K Carlson[7], Lindsay NK Davidson[1,2], Sonja V Fordham[8], Malcolm P Francis[9], Caroline M Pollock[10], Colin A Simpfendorfer[11,12], George H Burgess[13], Kent E Carpenter[14,15], Leonard JV Compagno[16], David A Ebert[17], Claudine Gibson[3], Michelle R Heupel[18], Suzanne R Livingstone[19], Jonnell C Sanciangco[14,15], John D Stevens[20], Sarah Valenti[3], William T White[20]

[1]IUCN Species Survival Commission Shark Specialist Group, Department of Biological Sciences, Simon Fraser University, Burnaby, Canada; [2]Earth to Ocean Research Group, Department of Biological Sciences, Simon Fraser University, Burnaby, Canada; [3]IUCN Species Survival Commission Shark Specialist Group, NatureBureau International, Newbury, United Kingdom; [4]Virginia Institute of Marine Science, College of William and Mary, Gloucester Point, United States; [5]British Antarctic Survey, Natural Environment Research Council, Cambridge, United Kingdom; [6]Research Institute for the Environment and Livelihoods, Charles Darwin University, Darwin, Australia; [7]Southeast Fisheries Science Center, NOAA/National Marine Fisheries Service, Panama City, United States; [8]Shark Advocates International, The Ocean Foundation, Washington, DC, United States; [9]National Institute of Water and Atmospheric Research, Wellington, New Zealand; [10]Global Species Programme, International Union for the Conservation of Nature, Cambridge, United Kingdom; [11]Centre for Sustainable Tropical Fisheries and Aquaculture, James Cook University, Townsville, Australia; [12]School of Earth and Environmental Sciences, James Cook University, Townsville, Australia; [13]Florida Program for Shark Research, Florida Museum of Natural History, University of Florida, Gainsville, United States; [14]IUCN Species Programme Species Survival Commission, Old Dominion University, Norfolk, United States; [15]Conservation International Global Marine Species Assessment, Old Dominion University, Norfolk, United States; [16]Shark Research Center, Iziko, South African Museum, Cape Town, South Africa; [17]Pacific Shark Research Center, Moss Landing Marine Laboratories, Moss Landing, United States; [18]School of Earth and Environmental Sciences, Australian Institute of Marine Science, Townsville, Australia; [19]Global Marine Species Assessment, Biodiversity Assessment Unit, IUCN Species Programme, Conservation International, Arlington, United States; [20]Marine and Atmospheric Research, Commonwealth Scientific and Industrial Research Organisation, Hobart, Australia

*For correspondence: dulvy@sfu.ca

Competing interests: The authors declare that no competing interests exist.

**Abstract** The rapid expansion of human activities threatens ocean-wide biodiversity. Numerous marine animal populations have declined, yet it remains unclear whether these trends are symptomatic of a chronic accumulation of global marine extinction risk. We present the first systematic analysis of threat for a globally distributed lineage of 1,041 chondrichthyan fishes—sharks, rays, and chimaeras. We estimate that one-quarter are threatened according to IUCN Red List criteria due to overfishing (targeted and incidental). Large-bodied, shallow-water species are at greatest risk and five out of the seven most threatened families are rays. Overall chondrichthyan extinction risk is substantially higher than for most other vertebrates, and only one-third of species are considered safe. Population depletion has occurred throughout the world's ice-free waters, but is particularly prevalent in the Indo-Pacific Biodiversity Triangle and Mediterranean Sea. Improved management of fisheries and trade is urgently needed to avoid extinctions and promote population recovery.

**eLife digest** Ocean ecosystems are under pressure from overfishing, climate change, habitat destruction and pollution. These pressures have led to documented declines of some fishes in some places, such as those living in coral reefs and on the high seas. However, it is not clear whether these population declines are isolated one-off examples or, instead, if they are sufficiently widespread to risk the extinction of large numbers of species.

Most fishes have a skeleton that is made of bone, but sharks and rays have a skeleton that is made of cartilage. A total of 1,041 species has such a skeleton and they are collectively known as the Chondrichthyes. To find out how well these fish are faring, Dulvy et al. worked with more than 300 scientists around the world to assess the conservation status of all 1,041 species.

Based on this, Dulvy et al. estimate that one in four of these species are threatened with extinction, mainly as a result of overfishing. Moreover, just 389 species (37.4% of the total) are considered to be safe, which is the lowest fraction of safe species among all vertebrate groups studied to date.

The largest sharks and rays are in the most peril, especially those living in shallow waters that are accessible to fisheries. A particular problem is the 'fin trade': the fins of sharks and shark-like rays are a delicacy in some Asian countries, and more than half of the chondrichthyans that enter the fin trade are under threat. Whether targeted or caught by boats fishing for other species, sharks and rays are used to supply a market that is largely unmonitored and unregulated. Habitat degradation and loss also pose considerable threats, particularly for freshwater sharks and rays.

Dulvy et al. identified three main hotspots where the biodiversity of sharks and rays was particularly seriously threatened—the Indo-Pacific Biodiversity Triangle, Red Sea, and the Mediterranean Sea— and argue that national and international action is needed to protect them from overfishing.

## Introduction

Populations and species are the building blocks of the communities and ecosystems that sustain humanity through a wide range of services (*Mace et al., 2005*; *Díaz et al., 2006*). There is increasing evidence that human impacts over the past 10 millennia have profoundly and permanently altered biodiversity on land, especially of vertebrates (*Schipper et al., 2008*; *Hoffmann et al., 2010*). The oceans encompass some of the earth's largest habitats and longest evolutionary history, and there is mounting concern for the increasing human influence on marine biodiversity that has occurred over the past 500 years (*Jackson, 2010*). So far our knowledge of ocean biodiversity change is derived mainly from retrospective analyses usually limited to biased subsamples of diversity, such as: charismatic species, commercially-important fisheries, and coral reef ecosystems (*Carpenter et al., 2008*; *Collette et al., 2011*; *McClenachan et al., 2012*; *Ricard et al., 2012*). Notwithstanding the limitations of these biased snapshots, the rapid expansion of fisheries and globalized trade are emerging as the principal drivers of coastal and ocean threat (*Polidoro et al., 2008*; *Anderson et al., 2011b*; *McClenachan et al., 2012*). The extent and degree of the global impact of fisheries upon marine biodiversity, however, remains poorly understood and highly contentious. Recent insights from ecosystem models and fisheries stock assessments of mainly data-rich northern hemisphere seas, suggest that the status of a few of the best-studied exploited species and ecosystems may be improving (*Worm et al., 2009*). However, this view is based on only 295 populations of 147 fish species and hence is far from representative of the majority of the world's fisheries and fished species, especially in the tropics for which there are few data and often less management (*Sadovy, 2005*; *Newton et al., 2007*; *Branch et al., 2011*; *Costello et al., 2012*; *Ricard et al., 2012*).

Overfishing and habitat degradation have profoundly altered populations of marine animals (*Hutchings, 2000*; *Lotze et al., 2006*; *Polidoro et al., 2012*), especially sharks and rays (*Stevens et al., 2000*; *Simpfendorfer et al., 2002*; *Dudley and Simpfendorfer, 2006*; *Ferretti et al., 2010*). It is not clear, however, whether the population declines of globally distributed species are locally reversible or symptomatic of an erosion of resilience and chronic accumulation of global marine extinction risk (*Jackson, 2010*; *Neubauer et al., 2013*). In response, we evaluate the scale and intensity of overfishing through a global systematic evaluation of the relative extinction risk for an entire lineage of exploited marine fishes—sharks, rays, and chimaeras (class Chondrichthyes)—using the Red List Categories and Criteria of the International Union for the Conservation of Nature (IUCN). We go on to identify, (i) the life

history and ecological attributes of species (and taxonomic families) that render them prone to extinction, and (ii) the geographic locations with the greatest number of species of high conservation concern.

Chondrichthyans make up one of the oldest and most ecologically diverse vertebrate lineages: they arose at least 420 million years ago and rapidly radiated out to occupy the upper tiers of aquatic food webs (*Compagno, 1990*; *Kriwet et al., 2008*). Today, this group is one of the most speciose lineages of predators on earth that play important functional roles in the top-down control of coastal and oceanic ecosystem structure and function (*Ferretti et al., 2010*; *Heithaus et al., 2012*; *Stevens et al., 2000*). Sharks and their relatives include some of the latest maturing and slowest reproducing of all vertebrates, exhibiting the longest gestation periods and some of the highest levels of maternal investment in the animal kingdom (*Cortés, 2000*). The extreme life histories of many chondrichthyans result in very low population growth rates and weak density-dependent compensation in juvenile survival, rendering them intrinsically sensitive to elevated fishing mortality (*Musick, 1999b*; *Cortés, 2002*; *García et al., 2008*; *Dulvy and Forrest, 2010*).

Chondrichthyans are often caught as incidental, but are often retained as valuable bycatch of fisheries that focus on more productive teleost fish species, such as tunas or groundfishes (*Stevens et al., 2005*). In many cases, fishing pressure on chondrichthyans is increasing as teleost target species become less accessible (due to depletion or management restrictions) and because of the high, and in some cases rising, value of their meat, fins, livers, and/or gill rakers (*Fowler et al., 2002*; *Clarke et al., 2006*; *Lack and Sant, 2009*). Fins, in particular, have become one of the most valuable seafood commodities: it is estimated that the fins of between 26 and 73 million individuals, worth US$400-550 million, are traded each year (*Clarke et al., 2007*). The landings of sharks and rays, reported to the Food and Agriculture Organization of the United Nations (FAO), increased steadily to a peak in 2003 and have declined by 20% since (*Figure 1A*). True total catch, however, is likely to be 3–4 times greater than reported (*Clarke et al., 2006*; *Worm et al., 2013*). Most chondrichthyan catches are unregulated and often misidentified, unrecorded, aggregated, or discarded at sea, resulting in a lack of species-specific landings information (*Barker and Schluessel, 2005*; *Clarke et al., 2006*; *Iglésias et al., 2010*; *Bornatowski et al., 2013*). Consequently, FAO could only be 'hopeful' that the catch decline is due to improved management rather than being symptomatic of worldwide overfishing (*FAO, 2010*). The reported chondrichthyan catch has been increasingly dominated by rays, which have made up greater than half of reported taxonomically-differentiated landings for the past four decades (*Figure 1B*). Chondrichthyan landings were worth US$1 billion at the peak catch in 2003, since then the value has dropped to US$800 million as catch has declined (*Musick and Musick, 2011*). A main driver of shark fishing is the globalized trade to meet Asian demand for shark fin soup, a traditional and usually expensive Chinese dish. This particularly lucrative trade in fins (not only from sharks, but also of shark-like rays such as wedgefishes and sawfishes) remains largely unregulated across the 86 countries and territories that exported >9,500 mt of fins to Hong Kong (a major fin trade hub) in 2010 (*Figure 1C*).

## Results

### Red List status of chondrichthyan species

Overall, we estimate that one-quarter of chondrichthyans are threatened worldwide, based on the observed threat level of assessed species combined with a modeled estimate of the number of Data Deficient species that are likely to be threatened. Of the 1,041 assessed species, 181 (17.4%) are classified as threatened: 25 (2.4%) are assessed as Critically Endangered (CR), 43 (4.1%) Endangered (EN), and 113 (10.9%) Vulnerable (VU) (*Table 1*). A further 132 species (12.7%) are categorized as Near Threatened (NT). Chondrichthyans have the lowest percentage (23.2%, n = 241 species) of Least Concern (LC) species of all vertebrate groups, including the marine taxa assessed to date (*Hoffmann et al., 2010*). Almost half (46.8%, n = 487) are Data Deficient (DD) meaning that information is insufficient to assess their status (*Table 1*). DD chondrichthyans are found across all habitats, but particularly on continental shelves (38.4% of 482 species in this habitat) and deepwater slopes (57.6%, *Table 2*). Of the 487 DD species for which we had sufficient maximum body size (n = 396) and geographic distribution data (n = 378), we were able to predict that at least a further 68 DD species are likely to be threatened (*Table 3*, *Supplementary file 1*). Accounting for the uncertainty in threat levels due to the number of DD species, we estimate that more than half face some elevated risk: at least one-quarter (n = 249; 24%) of chondrichthyans are threatened and well over one-quarter are Near Threatened (*Table 1*). Only 37% are predicted to be Least Concern (*Table 1*).

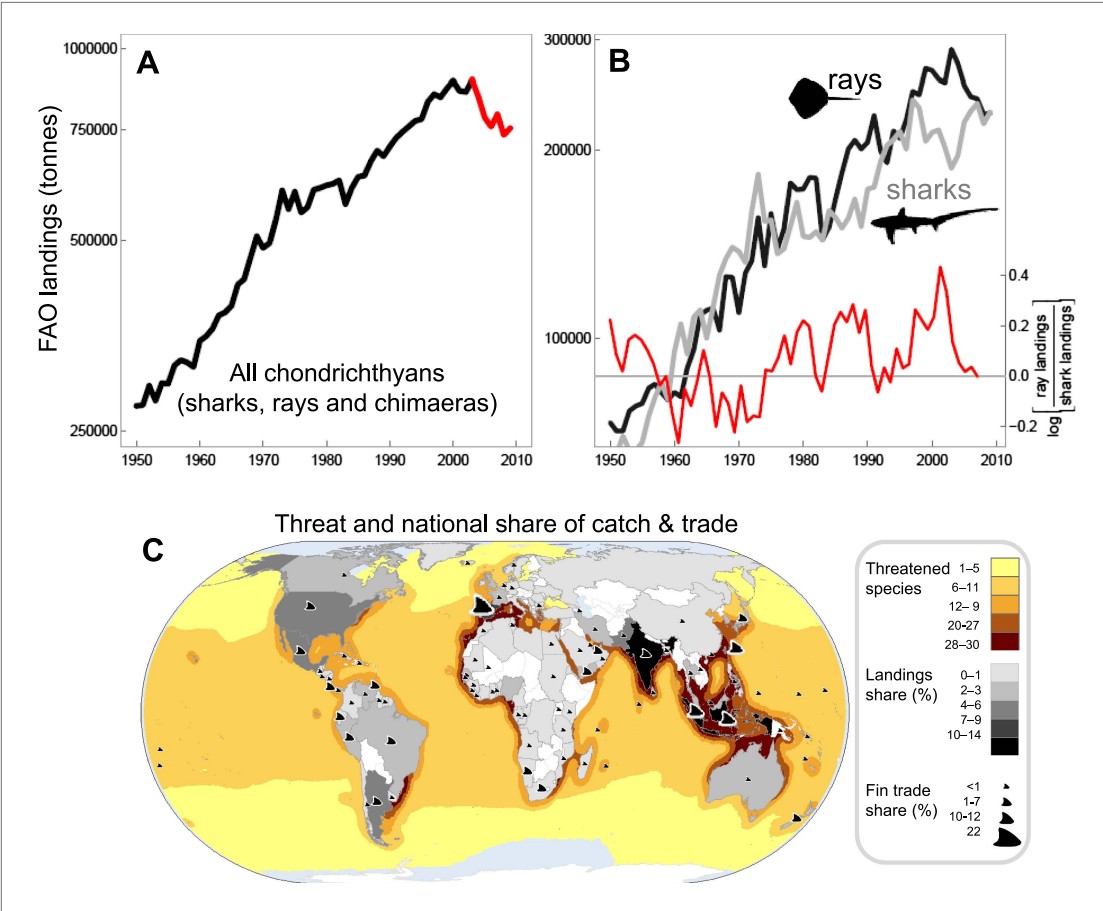

**Figure 1**. The trajectory and spatial pattern of chondrichthyan fisheries catch landings and fin exports. (**A**) The landed catch of chondrichthyans reported to the Food and Agriculture Organization of the United Nations from 1950 to 2009 up to the peak in 2003 (black) and subsequent decline (red). (**B**) The rising contribution of rays to the taxonomically-differentiated global reported landed catch: shark landings (light gray), ray landings (black), log ratio [rays/sharks], (red). Log ratios >0 occur when more rays are landed than sharks. The peak catch of taxonomically-differentiated rays peaks at 289,353 tonnes in 2003. (**C**) The main shark and ray fishing nations are gray-shaded according to their percent share of the total average annual chondrichthyan landings reported to FAO from 1999 to 2009. The relative share of shark and ray fin trade exports to Hong Kong in 2010 are represented by fin size. The taxonomically-differentiated proportion excludes the 'nei' (not elsewhere included) and generic 'sharks, rays, and chimaeras' category.

## Drivers of threat

The main threats to chondrichthyans are overexploitation through targeted fisheries and incidental catches (bycatch), followed by habitat loss, persecution, and climate change. While one-third of threatened sharks and rays are subject to targeted fishing, some of the most threatened species (including sawfishes and large-bodied skates) have declined due to incidental capture in fisheries targeting other species. Shark-like rays, especially sawfishes, wedgefishes and guitarfishes, have some of the most valuable fins and are highly threatened. Although the global fin trade is widely recognized as a major driver of shark and ray mortality, demand for meat, liver oil, and even gillrakers (of manta and other devil rays) also poses substantial threats. Half of the 69 high-volume or high-value sharks and rays in the global fin trade are threatened (53.6%, n = 37), while low-value fins often enter trade as well, even if meat demand is the main fishery driver (***Supplementary file 2A***). Coastal species are more exposed to the combined threats of fishing and habitat degradation than those offshore in pelagic and deepwater ecosystems. In coastal, estuarine, and riverine habitats, four principal processes of habitat degradation (residential and commercial development, mangrove destruction, river engineering, and pollution) jeopardize nearly one-third of threatened sharks and rays (29.8%, n = 54 of 181, ***Supplementary file 2B***). The combined effects of overexploitation and habitat degradation are most acute in freshwater, where over one-third (36.0%) of the 90 obligate and euryhaline freshwater chondrichthyans are threatened.

**Table 1.** Observed and predicted number and percent of chondrichthyan species in IUCN Red List categories

| Taxon | Species number (%) | Threatened species number (%) | CR | EN | VU | NT | LC | DD |
|---|---|---|---|---|---|---|---|---|
| Skates and rays | 539 (51.8) | 107 (19.9) | 14 (1.3) | 28 (2.7) | 65 (6.2) | 62 (6.0) | 114 (11.0) | 256 (24.6) |
| Sharks | 465 (44.7) | 74 (15.9) | 11 (1.1) | 15 (1.4) | 48 (4.6) | 67 (6.4) | 115 (11.0) | 209 (20.1) |
| Chimaeras | 37 (3.6) | 0 | 0 | 0 | 0 | 3 (0.3) | 12 (1.2) | 22 (2.1) |
| All observed | 1041 | 181 (17.4) | 25 (2.4) | 43 (4.1) | 113 (10.9) | 132 (12.7) | 241 (23.2) | 487 (46.8) |
| All predicted | | 249 (23.9) | – | – | – | 312 (29.9) | 389 (37.4) | 91 (8.7) |

CR, Critically Endangered; EN, Endangered; VU, Vulnerable; NT, Near Threatened; LC, Least Concern; DD, Data Deficient. Number threatened is the sum total of the categories CR, EN and VU. Species number and number threatened are expressed as percentage of the taxon, whereas the percentage of each species in IUCN categories is expressed relative to the total number of species.

Their plight is exacerbated by high habitat-specificity and restricted geographic ranges (**Stevens et al., 2005**). Specifically, the degradation of coastal, estuarine and riverine habitats threatened 14% of sharks and rays: through residential and commercial development (22 species, including river sharks *Glyphis* spp.); mangrove destruction for shrimp farming in Southeast Asia (4 species, including Bleeker's variegated stingray *Himantura undulata*); dam construction and water control (8 species, including Mekong freshwater stingray *Dasyatis laosensis*), and pollution (20 species). Many freshwater sharks and rays suffer multiple threats and have narrow geographic distributions, for example the Endangered Roughnose stingray *(Pastinachus solocirostris)* that is found only in Malaysian Borneo and Indonesia (Kalimantan, Sumatra and Java). Population control of sharks, in particular due to their perceived risk to people, fishing gear, and other fisheries has contributed to the threatened status of at least 12 species (**Supplementary file 2B**). Sharks and rays are also threatened due to capture in shark control nets (e.g. Dusky shark *Carcharhinus obscurus*), and persecution to minimise: damage to fishing nets (e.g. Green sawfish *Pristis zijsron*); their predation on aquacultured molluscs (e.g. Estuary stingray *Dasyatis fluviorum*); interference with spearfishing activity (e.g. Grey nurse shark *Carcharias taurus*), and the risk of shark attack (e.g. White shark *Carcharodon carcharias*). So far the threatened status of only one species has been directly linked to climate change (New Caledonia catshark *Aulohalaelurus kanakorum*, **Supplementary file 2B**). the climate-sensitivity of some sharks has been recognized (**Chin et al., 2010**) and the status of shark and ray species will change rapidly in climate cul-de-sacs, such as the Mediterranean Sea (**Lasram et al., 2010**).

## Correlates and predictors of threat

Elevated extinction risk in sharks and rays is a function of exposure to fishing mortality coupled with their intrinsic life history and ecological sensitivity (**Figures 2–6**). Most threatened chondrichthyan species are found in depths of less than 200 m, especially in the Atlantic and Indian Oceans, and the Western Central Pacific Ocean (79.6%, n = 144 of 181, **Figure 2**). Extinction risk is greater in larger-bodied

**Table 2.** Number and percent of chondrichthyans in IUCN Red List categories by their main habitats

| Habitat | Species (%) | Threatened (%) | CR (%) | EN (%) | VU (%) | NT (%) | LC (%) | DD (%) |
|---|---|---|---|---|---|---|---|---|
| Coastal and continental shelf | 482 (46.3) | 127 (26.3) | 20 (4.1) | 26 (5.4) | 81 (16.8) | 73 (15.1) | 97 (20.1) | 185 (38.4) |
| Neritic and epipelagic | 39 (3.7) | 17 (43.6) | 0 | 3 (7.7) | 14 (35.9) | 13 (33.3) | 5 (12.8) | 4 (10.3) |
| Deepwater | 479 (46.0) | 25 (5.2) | 2 (0.4) | 6 (1.3) | 17 (3.5) | 45 (9.4) | 133 (27.8) | 276 (57.6) |
| Mesopelagic | 8 (0.8) | 0 | 0 | 0 | 0 | 0 | 4 (50.0) | 4 (50.0) |
| Freshwater (obligate species only) | 33 (3.2) | 12 (36.4) | 3 (9.1) | 8 (24.2) | 1 (3.0) | 1 (3.0) | 2 (6.1) | 18 (54.5) |
| Totals | 1041 | 181 (17.4) | 25 (2.4) | 43 (4.1) | 113 (10.9) | 132 (12.7) | 241 (23.2) | 487 (46.8) |

CR, Critically Endangered; EN, Endangered; VU, Vulnerable; NT, Near Threatened; LC, Least Concern; DD, Data Deficient.

**Table 3.** Summary of predictive Generalized Linear Models for life history and ecological correlates of IUCN status

| Model | Model structure and hypothesis | Degrees of freedom, $k$ | Log likelihood | $AIC_c$ | $\Delta AIC$ | AIC weight | Accuracy (AUC) | $R^2$ |
|---|---|---|---|---|---|---|---|---|
| 1 | ~maximum length | 2 | −227.479 | 459 | 43.67 | 0.000 | 0.678 | 0.139 |
| 2 | ~ …+ minimum depth | 3 | −210.299 | 426.7 | 11.34 | 0.003 | 0.746 | 0.243 |
| 3 | ~ …+…+ depth range | 4 | −204.703 | 417.5 | 2.19 | 0.25 | 0.762 | 0.276 |
| 4 | ~ …+…+…+ geographic range | 5 | −202.578 | 415.3 | 0 | 0.748 | 0.772 | 0.298 |

Species were scored as threatened (CR, EN, VU) = 1 or Least Concern (LC) = 0 for n = 367 marine species. $AIC_c$ is the Akaike information criterion corrected for small sample sizes and $\Delta AIC$ is the change in $AIC_c$. The models are ordered by increasing complexity and decreasing AIC weight (largest $\Delta AIC$ to lowest), coefficient of determination ($R^2$), and prediction accuracy (measured using Area Under the Curve, AUC).

species found in shallower waters with narrower depth distributions, after accounting for phylogenetic non-independence (*Figures 3 and 4*). The traits with the greatest relative importance (>0.95) are maximum body size, minimum depth, and depth range. In comparison, geographic range (measured as Extent of Occurrence) has a much lower relative importance (0.79, *Figure 3*), and in the predictive models it improved the variance explained by 2% and the prediction accuracy by 1% (*Table 3*). The probability that a species is threatened increases by 1.2% for each 10 cm increase in maximum body length, and decreases by 10.3% for each 50 m deepening in the minimum depth limit of species. After accounting for maximum body size and minimum depth, species with narrower depth ranges have a 1.2% greater threat risk per 100 m narrowing of depth range. There is no significant interaction between depth range and minimum depth limit. Geographic range, measured as the Extent of Occurrence, varies over six orders of magnitude, between 354 km² and 278 million km² and is positively correlated with body size (Spearman's $\rho$ = 0.58), and hence is only marginally positively related to extinction risk over and above the effect of body size. Accounting for the body size and depth effects, the threat risk increases by only 0.5% for each 1,000,000 km² increase in geographic range (*Table 4*). The explanatory and predictive power of our life history and geographic distribution models increased with complexity, though geographic range size contributed relatively little additional explanatory power and a high degree of uncertainty in the parameter estimate (*Tables 3 and 4*). The maximum variance explained was 69% (*Table 4*) and the predictive models (without controlling for phylogeny) explained 30% of the variance and prediction accuracy was 77% (*Table 3*).

By habitat, one-quarter of coastal and continental shelf chondrichthyans (26.3%, n = 127 of 482) and almost half of neritic and epipelagic species (43.6%, n = 17 of 39) are threatened. Coastal and continental shelf and pelagic species greater than 1 m total length have a more than 50% chance of being threatened, compared to ~12% risk for a similar-sized deepwater species (*Figure 5*). While deepwater chondrichthyans, due to their slow growth and lower productivity, are intrinsically more sensitive to overfishing than their shallow-water relatives (*García et al., 2008*; *Simpfendorfer and Kyne, 2009*) for a given body size they are less threatened—largely because they are inaccessible to most fisheries (*Figure 5*).

As a result of their high exposure to coastal shallow-water fisheries and their large body size, sawfishes (Pristidae) are the most threatened chondrichthyan family and arguably the most threatened family of marine fishes (*Figure 6*). Other highly threatened families include predominantly coastal and continental shelf-dwelling rays (wedgefishes, sleeper rays, stingrays, and guitarfishes), as well as angel sharks and thresher sharks; five of the seven most threatened families are rays. Least threatened families are comprised of relatively small-bodied species occurring in mesopelagic and deepwater habitats (lanternsharks, catsharks, softnose skates, shortnose chimaeras, and kitefin sharks, *Figure 6*, *Figure 6—source data 1*).

## Geographic hotspots of threat and conservation priority by habitat

Local species richness is greatest in tropical coastal seas, particularly along the Atlantic and Western Pacific shelves (*Figure 7A*). The greatest uncertainty, where the number of DD species is highest, is centered on four areas: (1) Caribbean Sea and Western Central Atlantic Ocean, (2) Eastern Central Atlantic Ocean, (3) Southwest Indian Ocean, and (4) the China Seas (*Figure 7B*). The megadiverse China Seas face the triple jeopardy of high threat in shallow waters (*Figure 7CD*), high species richness (*Figure 7A*), and a large number of threatened endemic species (*Figure 8*), combined with high risk due to high uncertainty in status (large number of DD species, *Figure 7B*). Whereas the distribution of

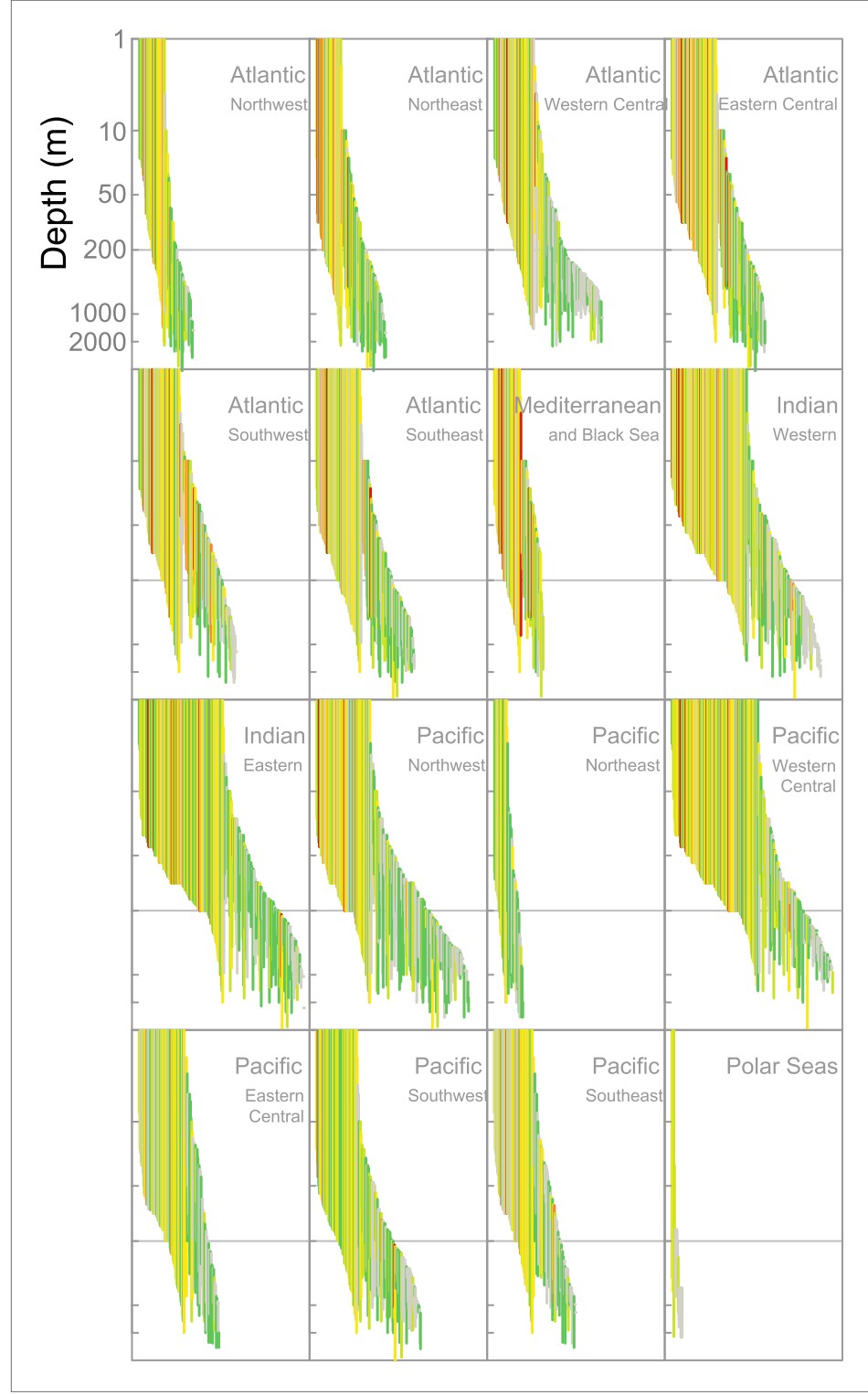

**Figure 2**. IUCN Red List Threat status and the depth distribution of chondrichthyans in the FAO Fishing Areas of the Atlantic, Indian and Pacific Oceans, and Polar Seas. Each vertical line represents the depth range (surface-ward minimum to the maximum reported depth) of each species and is colored according to threat status: CR (red), EN (orange), VU (yellow), NT (pale green), LC (green), and DD (gray). Species are ordered left to right by increasing median depth. The depth limit of the continental shelf is indicated by the horizontal gray line at 200 m. The Polar Seas include the following FAO Fishing Areas: Antarctic–Atlantic (Area 48), Indian (Area 58), Pacific (Area 88), and the Arctic Sea (Area 18).
*Figure 2. Continued on next page*

*Figure 2. Continued*
The following figure supplements are available for figure 2:
**Figure supplement 1**. Map of Food and Agriculture Organization of the United Nations Fishing Areas and their codes: 18, Arctic Sea; 21, Atlantic, Northwest; 27, Atlantic, Northeast; 31, Atlantic, Western Central; 34, Atlantic, Eastern Central; 37, Mediterranean and Black Sea; 41, Atlantic, Southwest; 47, Atlantic, Southeast; 48, Atlantic, Antarctic; 51, Indian Ocean, Western; 57, Indian Ocean, Eastern; 58, Indian Ocean, Antarctic and Southern; 61, Pacific, Northwest; 67, Pacific, Northeast; 71, Pacific, Western Central; 77, Pacific, Eastern Central; 81, Pacific, Southwest; 87, Pacific, Southeast; and, 88, Pacific, Antarctic.

threat in coastal and continental shelf chondrichthyans is similar to the overall threat pattern across tropical and mid-latitudes, the spatial pattern of threat varies considerably for pelagic and deepwater species. Threatened neritic and epipelagic oceanic sharks are distributed throughout the world's oceans, but there are also at least seven threat hotspots in coastal waters: (1) Gulf of California, (2) southeast US continental shelf, (3) Patagonian Shelf, (4) West Africa and the western Mediterranean Sea, (5) southeast South Africa, (6) Australia, and (7) the China Seas (*Figure 7D*). Hotspots of deepwater threatened chondrichthyans occur in three areas where fisheries penetrate deepest: (1) Southwest Atlantic Ocean (southeast coast of South America), (2) Eastern Atlantic Ocean, spanning from Norway to Namibia and into the Mediterranean Sea, and (3) southeast Australia (*Figure 7E*).

## Hottest hotspots of threat and priority

Spatial conservation priority can be assigned using three criteria: (1) the greatest number of threatened species (*Figure 7A*), (2) greater than expected threat (residuals of the relationship between total number of species and total number of threatened species per cell, *Figure 9*), and (3) high irreplaceability—high numbers of threatened endemic species (*Figure 8*). Most threatened marine chondrichthyans (n = 135 of 169) are distributed within, and are often endemic to (n = 73), at least seven distinct threat hotspots (e.g., for neritic and pelagic species *Figure 7D*). With the notable exception of the US and Australia, threat hotspots occur in the waters of the most intensive shark and ray fishing and fin-trading nations (*Figure 1C*). Accordingly these regions should be afforded high scientific and conservation priority (*Table 5*).

The greatest number of threatened species coincides with the greatest richness (*Figure 7A vs 7C–E*); by controlling for species richness we can reveal the magnitude of threat in the pelagic ocean and two coastal hotspots that have a greater than expected level of threat: the Indo-Pacific Biodiversity Triangle and the Red Sea. Throughout much of the pelagic ocean, threat is greater than expected based on species richness alone, species richness is low (n = 30) and a high percentage (86%) are threatened (n = 16) or Near Threatened (n = 10). Only four are of Least Concern (Salmon shark *Lamna ditropis*, Goblin shark *Mitsukurina owstoni*, Longnose pygmy Shark *Heteroscymnoides marleyi*, and Largetooth cookiecutter shark *Isistius plutodus*) (*Figure 9*). The Indo-Pacific Biodiversity Triangle, particularly the Gulf of Thailand, and the islands of Sumatra, Java, Borneo, and Sulawesi, is a hotspot of greatest residual threat especially for coastal sharks and rays with 76 threatened species (*Figure 9*). Indeed, the Gulf of Thailand large marine ecosystem has

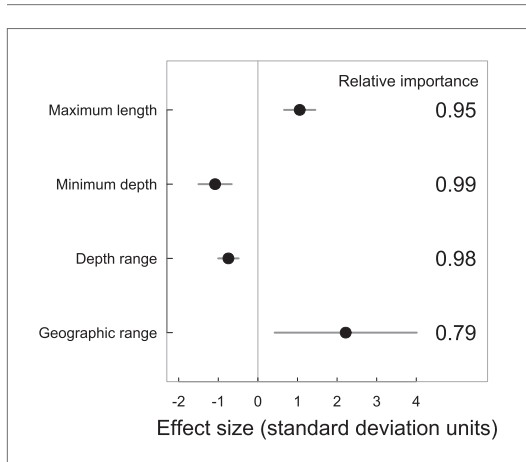

**Figure 3**. Standardized effect sizes with 95% confidence intervals from the two best explanatory models of life histories, geographic range and extinction risk in chondrichthyans. The data were standardized by subtracting the mean and dividing by one standard deviation to allow for comparison among parameters. The relative importance is calculated as the sum of the Akaike weights of the models containing each variable. Chondrichthyans were scored as threatened (CR, EN, VU) = 1 or Least Concern (LC) = 0 for n = 367 marine species. Threat status was modeled using General Linear Mixed-effects Models, with size, depth and geography treated as fixed effects and taxonomy hierarchy as a random effect to account for phylogenetic non-independence.

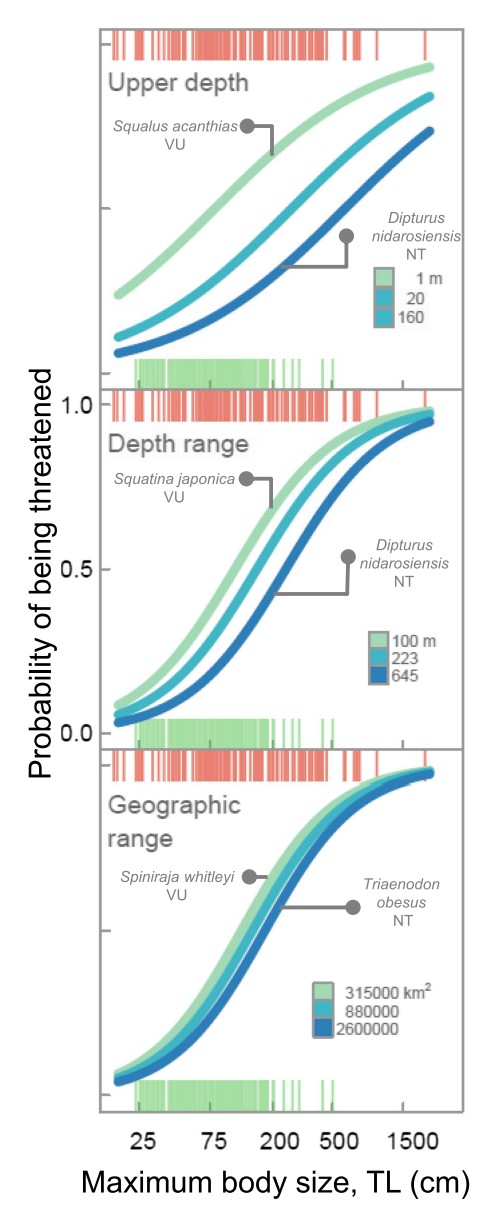

**Figure 4**. Life history sensitivity, accessibility to fisheries and extinction risk. Probability that a species is threatened due to the combination of intrinsic life history sensitivity (maximum body size, cm total length, TL) and accessibility to fisheries which is represented as minimum depth limit, depth range, and geographic range size (Extent of Occurrence). The lines represent the variation in body size-dependent risk for the upper quartile, median, and lower quartile of each range metric. The examplar species are all of similar maximum body length and the difference in risk is largely due to differences in geographic distribution. Chondrichthyans were scored as threatened (CR, EN, VU) = 1 or Least Concern (LC) = 0 for n = 366 marine species. The lines are the best fits from General Linear Mixed-effects Models, with maximum body size and geographic

*Figure 4. Continued on next page*

the highest threat density with 48 threatened chondrichthyans in an area of 0.36 million km². The Red Sea residual threat hotspot has 29 threatened pelagic and coastal species (*Figure 9*). There are 15 irreplaceable marine hotspots that harbor all 66 threatened endemic species (*Figure 8*; *Supplementary file 2C*).

## Discussion

In a world of limited funding, conservation priorities are often based on immediacy of extinction, the value of biodiversity and conservation opportunity (*Marris, 2007*). In this study, we provide the first estimates of the threat status and hence risk of extinction of chondrichthyans. Our systematic global assessment of the status of this lineage that includes many iconic predators reveals a risky combination of high threat (17% observed and 23.9% estimated), low safety (Least Concern, 23% observed and >37% estimated), and high uncertainty in their threat status (Data Deficient, 46% observed and 8.7% estimated). Over half of species are predicted to be threatened or Near Threatened (n = 561, 53.9%, *Table 1*). While no species has been driven to global extinction—as far as we know—at least 28 populations of sawfishes, skates, and angel sharks are locally or regionally extinct (*Dulvy et al., 2003*; *Dulvy and Forrest, 2010*). Several shark species have not been seen for many decades. The Critically Endangered Pondicherry shark (*Carcharhinus hemiodon*) is known only from 20 museum specimens that were captured in the heavily-fished inshore waters of Southeast Asia: it has not been seen since 1979 (*Cavanagh et al., 2003*). The now ironically-named and Critically Endangered Common skate (*Dipturus batis*) and Common angel shark (*Squatina squatina*) are regionally extinct from much of their former geographic range in European waters (*Cavanagh and Gibson, 2007*; *Gibson et al., 2008*; *Iglésias et al., 2010*). The Largetooth sawfish (*Pristis pristis*) and Smalltooth sawfish (*Pristis pectinata*) are possibly extinct throughout much of the Eastern Atlantic, particularly in West Africa (*Robillard and Séret, 2006*; *Harrison and Dulvy, 2014*).

Our analysis provides an unprecedented understanding of how many chondrichthyan species are actually or likely to be threatened. A very high percentage of species are DD (46%, 487 species); that is one of the highest rates of Data Deficiency of any taxon to date (*Hoffmann et al., 2010*). This high level of uncertainty in status further elevates risk and presents a key challenge for future assessment efforts. We outline a first step through our estimation that 68 DD species are likely to be

*Figure 4. Continued*

distribution traits treated as fixed effects and taxonomy hierarchy as a random effect to account for phylogenetic non-independence. Each vertical line in each of the 'rugs' represents the maximum body size and Red List status of each species: threatened (red) and LC (green).

threatened based on their life histories and distribution. Numerous studies have retrospectively explained extinction risk, but few have made a priori predictions of risk (*Dulvy and Reynolds, 2002*; *Davidson et al., 2012*). Across many taxa, extinction risk has been shown to be a function of an extrinsic driver or threat (*Jennings et al., 1998*; *Davies et al., 2006*) and the corresponding life history and ecological traits: large body size (low intrinsic rate of population increase, high trophic level), small geographic range size, and ecological specialization. Maximum body size is an essential predictor of threat status, we presume because of the close relationship between body size and the intrinsic rate of population increase in sharks and rays (*Smith et al., 1998*; *Frisk et al., 2001*; *Hutchings et al., 2012*). Though we note that this proximate link may be mediated ultimately through the time-related traits of growth and mortality (*Barnett et al., 2013*; *Juan-Jordá et al., 2013*). Our novel contribution is to show that depth-related geographic traits are more important for explaining risk than geographic range per se. The shallowness of species (minimum depth limit) and the narrowness of their depth range are important risk factors (*Figure 3*). We hypothesize that this is so because shallower species are more accessible to fishing gears and those with narrower depth ranges have lower likelihood that a proportion of the species distribution remains beyond fishing activity. For example, the Endangered Barndoor skate (*Dipturus laevis*) was eliminated throughout much of its geographic range and depth distribution due to bycatch in trawl fisheries, yet may have rebounded because a previously unknown deepwater population component lay beyond the reach of most fisheries (*Dulvy, 2000*; *Kulka et al., 2002*; *COSEWIC, 2010*). We find that geographic range (measured as Extent of Occurrence) is largely unrelated to extinction risk. This is in marked contrast to extinction risk patterns on land (*Jones et al., 2003*; *Cardillo et al., 2005*; *Anderson et al., 2011a*) and in the marine fossil record (*Harnik et al., 2012a*, *2012b*), where small geographic range size is the principal correlate of extinction risk. We suggest that this is because fishing activity is now widespread throughout the world's oceans (*Swartz et al., 2010*), and even species with the largest ranges are exposed and often entirely encompassed by the footprint of fishing activity. By contrast, with a few exceptions (mainly eastern Atlantic slopes, *Figure 7E*), fishing has a narrow depth penetration and hence species found at greater depths can still find refuge from exploitation (*Morato et al., 2006*; *Lam and Sadovy de Mitcheson, 2010*).

The status of chondrichthyans is arguably among the worst reported for any major vertebrate lineage considered thus far, apart from amphibians (*Stuart et al., 2004*; *Hoffmann et al., 2010*). The percentage and absolute number of threatened amphibians is high (>30% are threatened), but a greater percentage are Least Concern (38%), and uncertainty of status is lower (32% DD) than for chondrichthyans. Our discovery of the high level of threat in freshwater chondrichthyans (36%) is consistent with the emerging picture of the intense and unmanaged extinction risk faced by many freshwater and estuarine species (*Darwall et al., 2011*).

Our threat estimate is comparable to other marine biodiversity status assessments, but our findings caution that 'global' fisheries assessments may be underestimating risk. The IUCN Global Marine Species Assessment is not yet complete, but reveals varying threat levels among taxa and regions (*Polidoro et al., 2008*, *2012*). The only synoptic summary to-date focused on charismatic Indo-Pacific coral reef ecosystem species. Of the 1,568 IUCN-assessed marine vertebrates and invertebrates, 16% (range: 12–34% among families) were threatened (*McClenachan et al., 2012*). This is a conservative estimate of marine threat level because although they may be more intrinsically sensitive to extinction drivers, charismatic species are more likely to garner awareness of their status and support for monitoring and conservation (*McClenachan et al., 2012*). The predicted level of chondrichthyan threat (>24%) is distinctly greater than that provided by global fisheries risk assessments. These studies provide modeled estimates of the percentage of collapsed bony fish (teleost) stocks in both data-poor unassessed fisheries (18%, *Costello et al., 2012*) and data-rich fisheries (7–13%, *Branch et al., 2011*). This could be because teleosts are generally more resilient than elasmobranchs (*Hutchings et al., 2012*), but in addition we caution that analyses of biased geographic and taxonomic samples may be underestimating risk of collapse in global fisheries, particularly for species with less-resilient life histories.

Our work relies on consensus assessments by more than 300 scientists. However, given the uncertainty in some of the underlying data that inform our understanding of threat status, such as

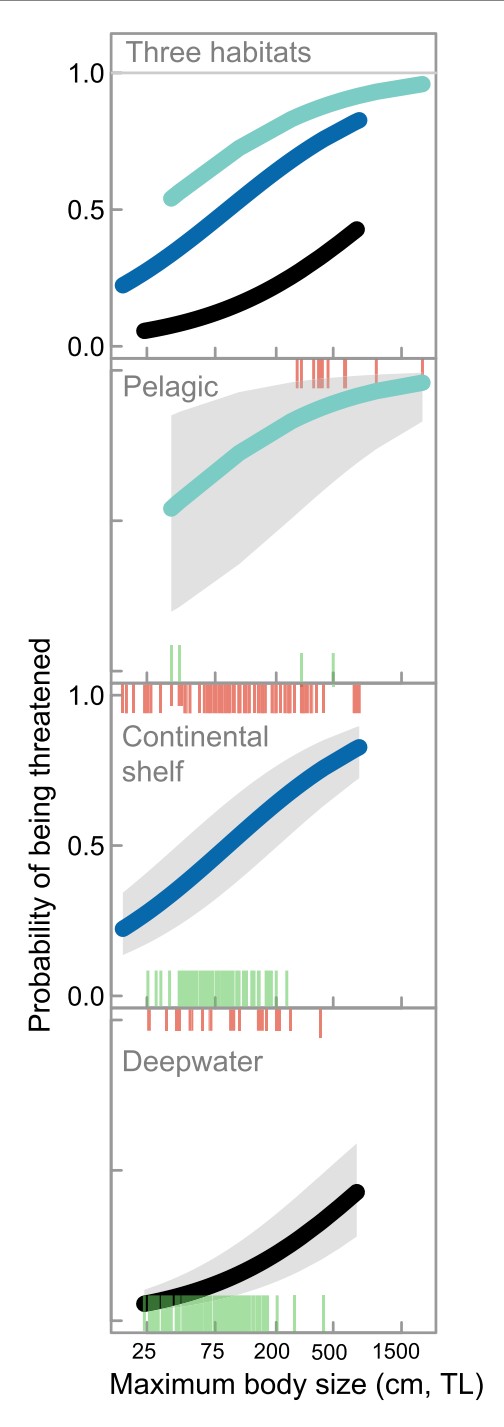

**Figure 5**. Life history, habitat, and extinction risk in chondrichthyans. IUCN Red List status as a function of maximum body size (total length, TL cm) and accessibility to fisheries in marine chondrichthyans in three main habitats: coastal and continental shelf <200 m ('Continental shelf'); neritic and oceanic pelagic <200 m ('Pelagic'); and, deepwater >200 m ('Deepwater'), n = 367 (threatened n = 148; Least Concern n = 219). The upper and lower 'rug' represents the maximum
*Figure 5. Continued on next page*

fisheries catch landings data, it is worth considering whether these uncertainties mean our assessments are downplaying the true risk. While there are methods of propagating uncertainty through the IUCN Red List Assessments (*Akcakaya et al., 2000*), in our experience this approach was uninformative for even the best-studied species, because it generated confidence intervals that spanned all IUCN Categories. Instead it is worth considering whether our estimates of threat are consistent with independent quantitative estimates of status. The Mediterranean Red List Assessment workshop in 2005 prompted subsequent quantitative analyses of catch landings, research trawl surveys, and sightings data. Quantitative trends could be estimated for five species suggesting they had declined by 96% to >99.9% relative to their former abundance suggesting they would meet the highest IUCN Threat category of Critically Endangered (*Ferretti et al., 2008*). By comparison the earlier IUCN regional assessment for these species, while suggesting they were all threatened, was more conservative for two of the five species: Hammerhead sharks (*Sphyrna* spp.)—Critically Endangered, Porbeagle shark (*Lamna nasus*)—Critically Endangered, Shortfin mako (*Isurus oxyrinchus*)—Critically Endangered, Blue shark (*Prionace glauca*)—Vulnerable, and Thresher shark (*Alopias vulpinus*)—Vulnerable.

We can also make a complementary comparison to a recent analysis of the status of 112 shark and ray fisheries (*Costello et al., 2012*). The median biomass relative to the biomass at Maximum Sustainable Yield (B/$B_{MSY}$) of these 112 shark and ray fisheries was 0.37, making them the most overfished groups of any of the world's unassessed fisheries. Assuming $B_{MSY}$ occurs at 0.3 to 0.5 of unexploited biomass then the median biomass of shark and ray fisheries had declined by between 81% and 89% by 2009. These biomass declines would be sufficient to qualify all of these 112 shark and ray fisheries for the Endangered IUCN category if they occurred within a three-generation time span. By comparison our results are considerably more conservative. Empirical analyses show that an IUCN threatened category listing is triggered only once teleost fishes (with far higher density-dependent compensation) have been fished down to below $B_{MSY}$ (*Dulvy et al., 2005*; *Porszt et al., 2012*). Hence, our findings are consistent with only around one-quarter of chondrichthyan species having been fished down below the $B_{MSY}$ target reference point. While there may be concern that expert assessments may overstate declines and threat, it is more likely that our con-

*Figure 5. Continued*

body size and Red List status of each species: threatened (upper rugs) and Least Concern (lower rugs). The lines are best fit using Generalized Linear Mixed-effects Models with 95% confidence intervals (*Table 9*).

servative consensus-based approach has understated declines and risk in sharks and rays.

For marine species, predicting absolute risk of extinction remains highly uncertain because, even with adequate evidence of severe decline, in many instances the absolute population size remains large (*Mace, 2004*). There remains considerable uncertainty as to the relationship between census and effective population size (*Reynolds et al., 2005*). Therefore, Red List categorization of chondrichthyans should be interpreted as a comparative measure of *relative* extinction risk, in recognition that unmanaged steep declines, even of large populations, may ultimately lead to ecosystem perturbations and eventually biological extinction. The Red List serves to raise red flags calling for conservation action, sooner rather than later, while there is a still chance of recovery and of forestalling permanent biodiversity loss.

Despite more than two decades of rising awareness of chondrichthyan population declines and collapses, there is still no global mechanism to ensure financing, implementation and enforcement of chondrichthyan fishery management plans that is likely to rebuild populations to levels where they would no longer be threatened (*Lack and Sant, 2009*; *Techera and Klein, 2011*). This management shortfall is particularly problematic given the large geographic range of many species. Threat increased only slightly when geographic range is measured as the Extent of Occurrence; however, geographic range becomes increasingly important when it is measured as the number of countries (legal jurisdictions) spanned by each species. The proportion of species that are threatened increases markedly with geographic size measured by number of Exclusive Economic Zones (EEZs) spanned; one-quarter of threatened species span the EEZs of 18 or more countries (*Figure 10*). Hence, their large geographic ranges do not confer safety, but instead exacerbates risk because sharks and rays require coherent, effective international management.

With a few exceptions (e.g., Australia and USA), many governments still lack the resources, expertise, and political will necessary to effectively conserve the vast majority of shark and rays, and indeed many

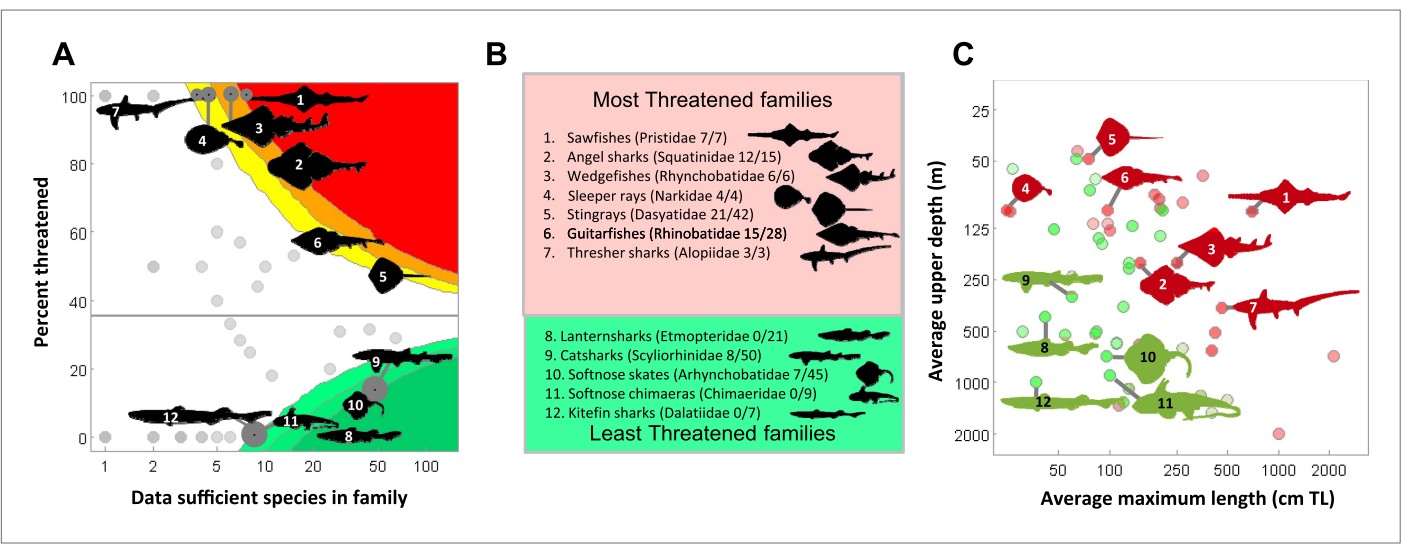

**Figure 6**. Evolutionary uniqueness and taxonomic conservation priorities. Threat among marine chondrichthyan families varies with life history sensitivity (maximum length) and exposure to fisheries (depth distribution). (**A**) Proportion of threatened data sufficient species and the richness of each taxonomic family. Colored bands indicate the significance levels of a one-tailed binomial test at p=0.05, 0.01, and 0.001. Those families with significantly greater (or lower) than expected threat levels at p<0.05 against a null expectation that extinction risk is equal across families (35.6%). (**B**) The most and least threatened taxonomic families. (**C**) Average life history sensitivity and accessibility to fisheries of 56 chondrichthyan families. Significantly greater (or lower) risk than expected is shown in red (green).
The following source data are available for figure 6:

**Source data 1**. Number and IUCN Red List status of chondrichthyan species in IUCN Red List categories by family (alphabetically within each order).

**Table 4.** Summary of explanatory Generalized Linear Mixed-effect Models of the life history and geographic distributional correlates of IUCN status

| Model structure and hypothesis | Degrees of freedom, $k$ | Log likelihood | $AIC_c$ | $\Delta AIC$ | AIC weight | $R^2$GLMM($m$) of fixed effects only | $R^2$GLMM($c$) of fixed and random effects |
|---|---|---|---|---|---|---|---|
| ~ maximum length | 5 | −197.06 | 404.3 | 28.31 | 0.000 | 0.32 | 0.58 |
| ~ …+ minimum depth | 6 | −187.013 | 386.3 | 10.29 | 0.005 | 0.48 | 0.65 |
| ~ …+…+ depth range | 7 | −182.139 | 378.6 | 2.62 | 0.212 | 0.49 | 0.66 |
| ~ …+…+…+ geographic range | 8 | −179.785 | 376.0 | 0 | 0.784 | 0.69 | 0.80 |

Species were scored as threatened (CR, EN, VU) = 1 or Least Concern (LC) = 0 for n = 367 marine species. $AIC_c$ is the Akaike information criterion corrected for small sample sizes; $\Delta AIC$ is the change in $AIC_c$. The models are ordered by increasing complexity and decreasing AIC weight (largest $\Delta AIC$ to lowest). $R^2$GLMM($m$) is the marginal $R^2$ of the fixed effects only and $R^2$GLMM($c$) is the conditional $R^2$ of the fixed and random effects.

other exploited organisms (*Veitch et al., 2012*). More than 50 sharks are included in Annex I (Highly Migratory Species) of the 1982 Law of the Sea Convention, implemented on the high seas under the 1992 Fish Stocks Agreement, but currently only a handful enjoy species-specific protections under the world's Regional Fishery Management Organizations (*Table 6*), and many of these have yet to be implemented domestically. The Migratory Sharks Memorandum of Understanding (MoU) adopted by the Parties to the Convention on Migratory Species (CMS) so far only covers seven sharks, yet there may be more than 150 chondrichthyans that regularly migrate across national boundaries (*Fowler, 2012*). To date, only one of the United Nations Environment Programme's Regional Seas Conventions, the Barcelona Convention for the Conservation of the Mediterranean Sea, includes chondrichthyan fishes and only a few of its Parties have taken concrete domestic action to implement these listings. Despite two decades of effort, only ten sharks and rays had been listed by the Convention on International Trade in Endangered Species (CITES) up to 2013 (*Vincent et al., 2014*). A further seven species of shark and ray were listed by CITES in 2013—the next challenge is to ensure effective implementation of these trade regulations (*Mundy-Taylor and Crook, 2013*). OSPAR (the Convention for the Protection of the marine Environment of the North-East Atlantic) lists many threatened shark and ray species, but its remit excludes fisheries issues. Many chondrichthyans qualify for listing under CITES, CMS, and various regional seas conventions, and should be formally considered for such action as a complement to action by Regional Fisheries Management Organizations (RFMOs) (*Table 6*).

Bans on 'finning' (slicing off a shark's fins and discarding the body at sea) are the most widespread shark conservation measures. While these prohibitions, particularly those that require fins to remain attached through landing, can enhance monitoring and compliance, they have not significantly reduced shark mortality or risk to threatened species (*Clarke et al., 2013*). Steep declines and the high threat levels in migratory oceanic pelagic sharks suggest raising the priority of improved management of catch and trade through concerted actions by national governments working through RFMOs *as well as* CITES, and CMS (*Table 7*).

A high proportion of catch landings come from nations with a large number of threatened chondrichthyans and less-than-comprehensive chondrichthyan fishery management plans. Future research is required to down-scale these global Red List assessments and analyses to provide country-by-country diagnoses of the link between specific fisheries and specific threats to populations of more broadly distributed species (*Wallace et al., 2010*). Such information could be used to focus fisheries management and conservation interventions that are tailored to specific problems. There is no systematic global monitoring of shark and ray populations and the national fisheries catch landings statistics provide invaluable data for tracking fisheries trends in unmanaged fisheries (*Newton et al., 2007*; *Worm et al., 2013*). However, the surveillance power of such data could be greatly improved if collected at greater taxonomic resolution. While there have been continual improvements, catches are underreported (*Clarke et al., 2006*), and for those that are reported only around one-third is reported at the

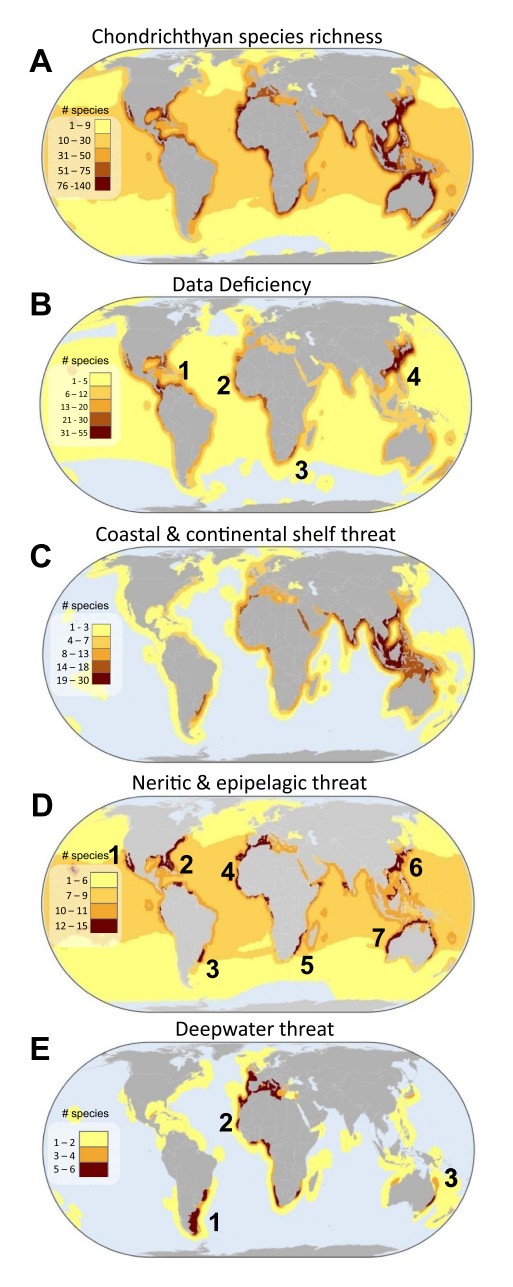

**Figure 7**. Global patterns of marine chondrichthyan diversity, threat and knowledge. (**A**) Total chondrichthyan richness, (**B**) the number of Data Deficient and threat by major habitat: (**C**) coastal and continental shelf (<200 m depth), (**D**) neritic and epipelagic (<200 m depth), and (**E**) deepwater slope and abyssal plain (>200 m) habitats. Numbers expressed as the total number of species in each 23,322 km² cell. The numbers are hotspots refereed to in the text.

species level (*Fischer et al., 2012*). To complement improved catch landings data, we recommend the development of repeat regional assessments of the Red List Status of chondrichthyans to provide an early warning of adverse changes in status and to detect and monitor the success of management initiatives and interventions. Aggregate Red List Threat indices for chondrichthyans, like those available for mammals, birds, amphibians, and hard corals (*Carpenter et al., 2008*) would provide one of the few global scale indicators of progress toward international biodiversity goals (*Walpole et al., 2009*; *Butchart et al., 2010*).

Our global status assessment of sharks and rays reveals the principal causes and severity of global marine biodiversity loss, and the threat level they face exposes a serious shortfall in the conservation management of commercially-exploited aquatic species (*McClenachan et al., 2012*). Chondrichthyans have slipped through the jurisdictional cracks of traditional national and international management authorities. Rather than accept that many chondrichthyans will inevitably be driven to economic, ecological, or biological extinction, we warn that dramatic changes in the enforcement and implementation of the conservation and management of threatened chondrichthyans are urgently needed to ensure a healthy future for these iconic fishes and the ecosystems they support.

## Methods

### IUCN Red List Assessment process and data collection

We applied the Red List Categories and Criteria developed by the International Union for Conservation of Nature (IUCN) (*IUCN, 2004*) to 1,041 species at 17 workshops involving more than 300 experts who incorporated all available information on distribution, catch, abundance, population trends, habitat use, life histories, threats, and conservation measures.

Some 105 chondrichthyan fish species had been assessed and published in the 2000 Red List of Threatened Species prior to the initiation of the Global Shark Red List Assessment (GSRLA). These assessments were undertaken by correspondence and through discussions at four workshops (1996—London, UK, and Brisbane, Australia; 1997—Noumea, New Caledonia, and 1999—Pennsylvania, USA). These assessments applied earlier versions of the IUCN Red List Criteria and, where possible, were subsequently reviewed and updated according to version 3.1 Categories and Criteria during the GSRLA. The IUCN Shark Specialist Group (SSG) subsequently held a series of 13 regional and thematic Red List workshops in nine countries around the world (*Table 8*). Prior to the workshops, each

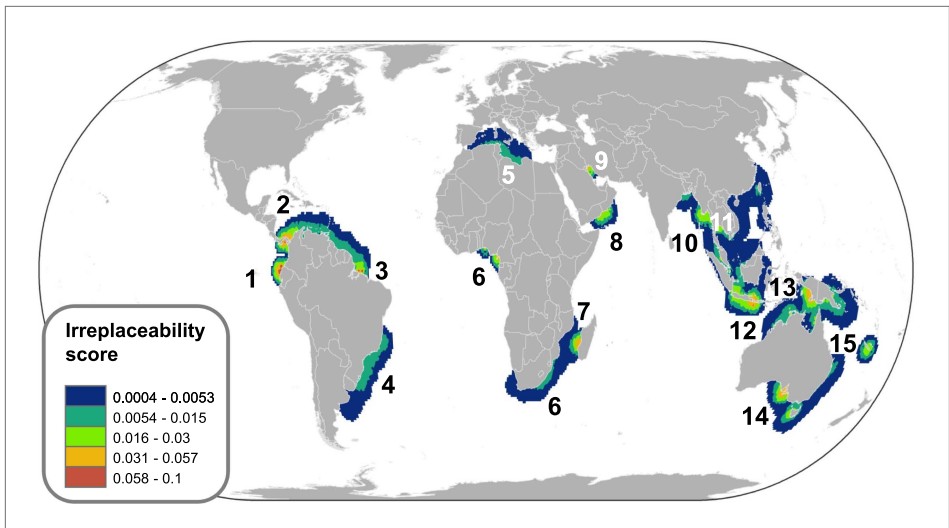

**Figure 8**. Irreplaceability hotspots of the endemic threatened marine chondrichthyans. Endemics were defined as species with an Extent of Occurrence of <500,000 km$^2$ (n = 66). Irreplaceable cells with the greatest number of small range species are shown in red, with blue cells showing areas of lower, but still significant irreplaceability. Irreplaceability is the sum of the inverse of the geographic range sizes of all threatened endemic species in the cell. A value of 0.1 means that on average a single cell represents one tenth of the global range of all the species present in the cell. The numbers are hotspots referred to in the text.

participant was asked to select species for assessment based on their expertise and research areas. Where possible, experts carried out research and preparatory work in advance, thus enabling more synthesis to be achieved during each workshop. SSG Red List-trained personnel facilitated discussion and consensus sessions, and coordinated the production of global Red List Assessments for species in each region. For species that had previously been assessed, participants provided updated information and assisted in revised assessments. Experts completed assessments for some wide-ranging, globally distributed species over the course of several workshops. In total, 302 national, regional, and international experts from 64 countries participated in the GSRLA workshops and the production of assessments. All Red List Assessments were based on the collective knowledge and pooled data from dedicated experts

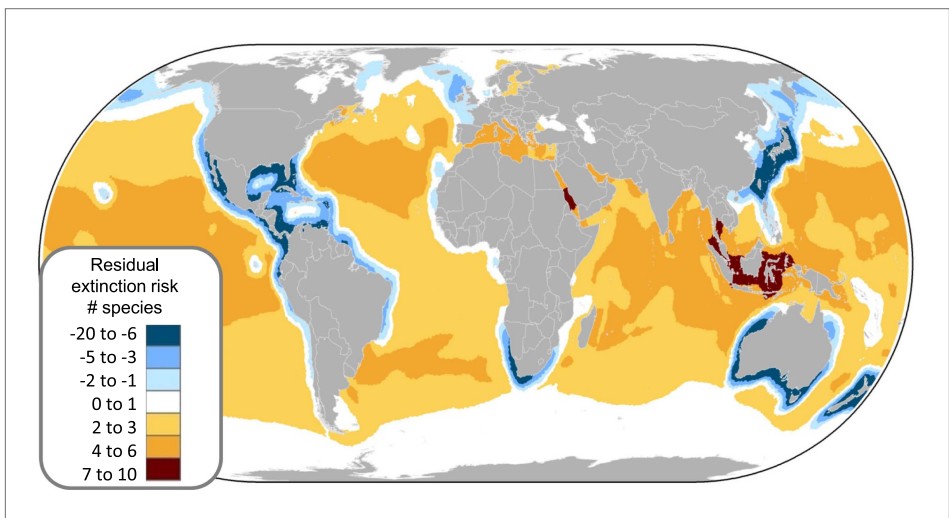

**Figure 9**. Spatial variation in the relative extinction risk of marine chondrichthyans. Residuals of the relationship between total number of data sufficient chondrichthyans and total number of threatened species per cell, where positive values (orange to red) represent cells with higher threat than expected for their richness alone.

**Table 5.** Scientific and conservation priority according to threat, knowledge and endemicity by FAO Fishing Area

| FAO Fishing Area (ranked priority) | Threatened species (% of total, n = 181) | Data Deficient species (% of total, n = 487) | Number of endemic species (threatened endemics) | Threatened endemic species |
|---|---|---|---|---|
| (1) Indian, Eastern | 67 (37.0) | 69 (14.2) | 58 (5) | Atelomycterus baliensis, Himantura fluviatilis, Zearaja maugeana, Trygonorrhina melaleuca, Urolophus orarius |
| (2) Pacific, Western Central | 76 (42.0) | 81 (16.6) | 51 (14) | Glyphis glyphis, Aulohalaelurus kanakorum, Hemitriakis leucoperiptera, Brachaelurus colcloughi, Hemiscyllium hallstromi, H. strahani, Himantura hortlei, H. lobistoma, Pastinachus solocirostris, Aptychotrema timorensis, Rhinobatos jimbaranensis, Rhynchobatus sp. nov. A, Rhynchobatus springeri, Urolophus javanicus |
| (3) Pacific, Northwest | 48 (26.5) | 116 (23.8) | 80 (6) | Benthobatis yangi, Narke japonica, Raja pulchra, Squatina formosa, S. japonica, S. nebulosa |
| (4) Indian, Western | 61 (33.7) | 104 (21.4) | 62 (8) | Carcharhinus leiodon, Haploblepharus kistnasamyi, H. favus, H. punctatus, Pseudoginglymostoma brevicaudatum, Electrolux addisoni, Dipturus crosnieri, Okamejei pita |
| (5) Atlantic, Western Central | 32 (17.7) | 81 (16.6) | 62 (4) | Diplobatis colombiensis, D. guamachensis, D. ommata, D. pictus |
| (6) Pacific, Southwest | 34 (18.8) | 49 (10.1) | 28 | |
| (7) Atlantic, Southwest | 52 (28.7) | 52 (10.7) | 37 (19) | Galeus mincaronei, Schroederichthys saurisqualus, Mustelus fasciatus, M. schmitti, Atlantoraja castelnaui, A. cyclophora, A. platana, Rioraja agassizii, Sympterygia acuta, Benthobatis kreffti, Dipturus mennii, Gurgesiella dorsalifera, Rhinobatos horkelii, Zapteryx brevirostris, Rhinoptera brasiliensis, Squatina argentina, S. guggenheim, S. occulta, S. punctata |
| (8) Atlantic, Southeast | 37 (20.4) | 51 (10.5) | 13 | |
| (9) Atlantic, Eastern Central | 42 (23.2) | 44 (9.0) | 6 | |
| (10) Pacific, Southeast | 26 (14.4) | 67 (13.8) | 32 (3) | Mustelus whitneyi, Triakis acutipinna, T. maculata |
| (11) Pacific, Eastern Central | 20 (11.0) | 52 (10.7) | 19 (2) | Urotrygon reticulata, U. simulatrix |
| (12) Atlantic, Northeast | 33 (18.2) | 23 (4.7) | 8 | |
| (13) Atlantic, Northwest | 22 (12.2) | 17 (3.5) | 3 (1) | Malacoraja senta |
| (14) Mediterranean & Black Sea | 34 (18.8) | 16 (3.3) | 3 (1) | Leucoraja melitensis |
| (15) Pacific, Northeast | 9 (5.0) | 11 (2.3) | 0 | |
| (16) Indian, Antarctic | 1 (0.6) | 4 (0.8) | 2 | |
| (17) Atlantic, Antarctic | 1 (0.6) | 4 (0.8) | 2 | |
| (18) Pacific, Antarctic | 0 | 3 (0.6) | 0 | |
| (19) Arctic Sea | 0 | 0 | 0 | |

Endemics were defined as those species found only within a single FAO Fishing Area. FAO Fishing Areas were ranked according to greatest species richness, percent threatened species, percent Data Deficient species, number of endemic species and number of threatened endemic species.

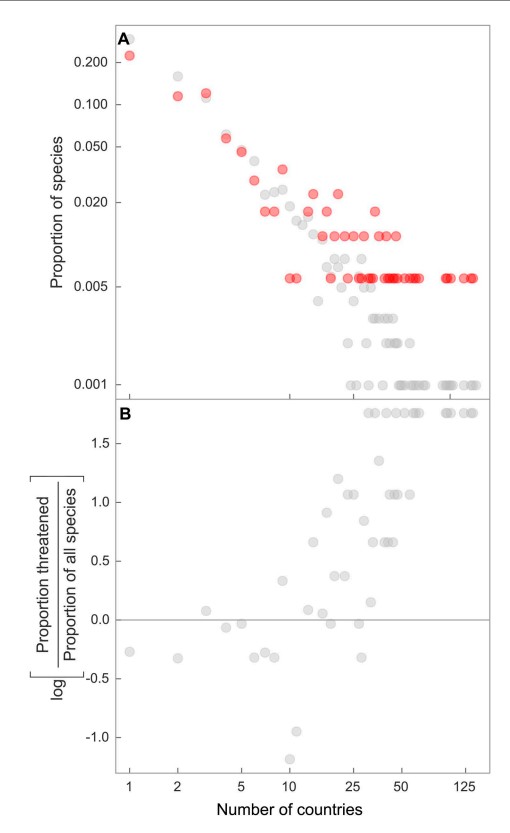

**Figure 10**. Elevated threat in chondrichthyans with the largest geographic ranges, spanning the greatest number of national jurisdictions. Frequency distribution of number of jurisdictions spanned by all chondrichthyans (black, n = 1,041) and threatened species only (red, n = 174), for (**A**) country EEZs, and (**B**) the overrepresentation of threatened species spanning a large number of country EEZs, shown by the log ratio of proportion of threatened species over the proportion of all species. The proportion of threatened species is greater than the proportion of all species where the log ratio = 0, which corresponds to range spans of 16 and more countries. DOI: 10.7554/eLife.00590.019

across the world, ensuring global consultation and consensus to achieve the best assessment for each species with the knowledge and resources available ('Acknowledgements'). Any species assessments not completed during the workshops were finalized through subsequent correspondence among experts.

The SSG evaluated the status of all described chondrichthyan species that are considered to be taxonomically valid up to August 2011 (see "Systematics, missing species and species coverage" below). Experts compiled peer-reviewed Red List documentation for each species, including data on: systematics, population trends, geographic range, habitat preferences, ecology, life history, threats, and conservation measures. The SSG assessed all species using the IUCN Red List Categories and Criteria version 3.1 (**IUCN, 2001**). The categories and their standard abbreviations are: Critically Endangered (CR), Endangered (EN), Vulnerable (VU), Near Threatened (NT), Least Concern (LC), and Data Deficient (DD). Experts further coded each species according to the IUCN Habitats, Threats and Conservation Actions Authority Files, enabling analysis of their habitat preferences, major threats and conservation action requirements. SSG Program staff entered all data into the main data fields in the IUCN Species Information Service Data Entry Module (SIS DEM) and subsequently transferred these data into the IUCN Species Information Service (SIS) in 2009.

## Systematics, missing species and species coverage

The SSG collated data on order, family, genus, species, taxonomic authority, commonly-used synonyms, English common names, other common names, and taxonomic notes (where relevant). For taxonomic consistency throughout the species assessments, the SSG followed Leonard J V Compagno's 2005 Global Checklist of Living Chondrichthyan Fishes (**Compagno, 2005**), only deviating from this where there was extensive opposing consensus with a clear and justifiable alternative, as adjudicated by the IUCN SSG's Vice Chairs of Taxonomy, David E Ebert and William T White.

Keeping pace with the total number of chondrichthyans is a challenging task, especially given the need to balance immediacy against taxonomic stability. One-third of all species have been described in the past thirty years. Scientists have described a new chondrichthyan species, on average, almost every 2–3 weeks since the 1970s (**Last, 2007**; **White and Last, 2012**). Since Leonard V J Compagno completed the global checklist in 2005, scientists have recognized an additional ~140 species (mostly new) living chondrichthyan species. This increase in the rate of chondrichthyan descriptions in recent years is primarily associated with the lead up to the publication of a revised treatment of the entire chondrichthyan fauna of Australia (**Last and Stevens, 2009**), requiring formal descriptions of previously undescribed taxa. In particular, three CSIRO special publications published in 2008 included descriptions of 70 previously undescribed species worldwide (**Last et al., 2008a**, **2008b**, **2008c**). The number of new species described in 2006, 2007 and 2008 was 21, 23, and 81, respectively, with all but

nine occurring in the Indo–West Pacific. Additional nominal species of chondrichthyans are also included following resurrection of previously unrecognized species such as the resurrection of *Pastinachus atrus* for the Indo–Australian region, previously considered a synonym of *P. sephen* (**Last and Stevens, 1994**). Scientists excluded some nominal species of dubious taxonomic validity from this assessment. Thus, the total number of chondrichthyan species referred to in this paper (1,041) does not include all recent new or resurrected species, which require future work for their inclusion in the GSRLA.

Many more as yet undescribed chondrichthyan species exist. The chondrichthyan faunas in several parts of the world (e.g., the northern Indian Ocean) are poorly known and a large number of species are likely to represent complexes of several distinct species that require taxonomic resolution, for example some dogfishes, skates, eagle rays, and stingrays (**Iglésias et al., 2010**; **White and Last, 2012**). Many areas of the Indian and Pacific Oceans are largely unexplored and, given the level of micro-endemism documented for a number of chondrichthyan groups, it is likely that many more species will be discovered in the future (**Last, 2007**; **Naylor et al., 2012**). For example, recent surveys of Indonesian fish markets revealed more than 20 new species of sharks out of the approximately 130 recorded in total (**White et al., 2006**; **Last, 2007**; **Ward et al., 2008**).

**Table 6.** Progress toward regional and international RFMO management measures for sharks and rays

1. Bans on 'finning' (the removal of a shark's fins and discarding the carcass at sea) through most RFMOs (**Fowler and Séret, 2010**);

2. North East Atlantic Fisheries Commission (NEAFC) bans on directed fishing for species not actually targeted within the relevant area (Spiny dogfish [*Squalus acanthias*], Basking shark [*Cetorhinus maximus*], Porbeagle shark [*Lamna nasus*]) (**NEAFC, 2009**);

3. Convention on the Conservation of Antarctic Marine Living Resources bans on 'directed' fishing for skates and sharks and bycatch limits for skates (**CCMLR, 2011**);

4. A Northwest Atlantic Fisheries Organization (NAFO) skate quota (note: this has consistently been set higher than the level advised by scientists since its establishment in 2004) (**NAFO, 2011**);

5. International Commission for the Conservation of Atlantic Tunas (ICCAT) bans on retention, transshipment, storage, landing, and sale of Bigeye Thresher (*Alopias superciliosus*), and Oceanic whitetip shark (*Carcharhinus longimanus*), and partial bans (developing countries excepted under certain circumstances) on retention, transshipment, storage, landing, and sale of most hammerheads (*Sphyrna* spp.), and retention, transshipment, storage, and landing (but not sale) of Silky shark (*Carcharhinus falciformis*) (**Kyne et al., 2012**);

6. An Inter-American Tropical Tuna Commission (IATTC) ban on retention, transshipment, storage, landing, and sale of Oceanic whitetip sharks (**IATTC, 2011**);

7. An Indian Ocean Tuna Commission (IOTC) ban on retention, transshipment, storage, landing, and sale of thresher sharks-with exceptionally low compliance and reportedly low effectiveness (**IOTC, 2011**); and,

8. A Western and Central Pacific Fisheries Commission ban on retention, transshipment, storage, and landing (but not sale) of Oceanic whitetip sharks (**Clarke et al., 2013**).

## Distribution maps

SSG experts created a shapefile of the geographic distribution for each chondrichthyan species with GIS software using the standard mapping protocol for marine species devised by the IUCN GMSA team (http://sci.odu.edu/gmsa/). The map shows the Extent of Occurrence of the species cut to one of several standardized basemaps depending on the ecology of the species (i.e., coastal and continental shelf, pelagic and deepwater). The distribution maps for sharks are based on original maps provided by the FAO and Leonard JV Compagno. Maps for some of the batoids were originally provided by John McEachran. New maps for recently described species were drafted where necessary. The original maps were updated, corrected, or verified by experts at the Red List workshops or out-of-session assessors and SSG staff and then sent to the GMSA team who modified the shapefiles and matched them to the distributional text within the assessment.

## Occurrence and habitat preference

SSG assessors assigned countries of occurrence from the 'geographic range' section of the Red List documentation and classified species to the FAO Fishing Areas (http://www.iucnredlist.org/technical-documents/data-organization) in which they occur (*Figure 2—figure supplement 1*). Each species was coded according to the IUCN Habitats Authority File (http://www.iucnredlist.org/technical-documents/classification-schemes/habitats-classification-scheme-ver3). These categorizations are poorly developed and often irrelevant for coastal and offshore marine animals. For the purposes of analysis presented here we assigned chondrichthyans to five unique habitat-lifestyle combinations (coastal and continental shelf, pelagic, meso- and bathypelagic, deepwater,

**Table 7.** Management recommendations: the following actions would contribute to rebuilding threatened chondrichthyan populations and properly managing associated fisheries

Fishing nations and regional fisheries management organizations (RFMOs) are urged to:

1. Implement, as a matter of priority, scientific advice for protecting habitat and/or preventing overfishing of chondrichthyan populations;

2. Draft and implement Plans of Action pursuant to the International Plan Of Action (IPOA–Sharks), which include, wherever possible, binding, science-based management measures for chondrichthyans and their essential habitats;

3. Significantly increase observer coverage, monitoring, and enforcement in fisheries taking chondrichthyans;

4. Require the collection and accessibility of species-specific chondrichthyan fisheries data, including discards, and penalize non-compliance;

5. Conduct population assessments for chondrichthyans;

6. Implement and enforce chondrichthyan fishing limits in accordance with scientific advice; when sustainable catch levels are uncertain, set limits based on the precautionary approach;

7. Strictly protect chondrichthyans deemed exceptionally vulnerable through Ecological Risk Assessments and those classified by IUCN as Critically Endangered or Endangered;

8. Prohibit the removal of shark fins while onboard fishing vessels and thereby require the landing of sharks with fins naturally attached; and,

9. Promote research on gear modifications, fishing methods, and habitat identification aimed at mitigating chondrichthyan bycatch and discard mortality.

National governments are urged to:

10. Propose and work to secure RFMO management measures based on scientific advice and the precautionary approach;

11. Promptly and accurately report species-specific chondrichthyan landings to relevant national and international authorities;

12. Take unilateral action to implement domestic management for fisheries taking chondrichthyans, including precautionary limits and/or protective status where necessary, particularly for species classified by IUCN as Vulnerable, Endangered or Critically Endangered, and encourage similar actions by other Range States;

13. Adopt bilateral fishery management agreements for shared chondrichthyan populations;

14. Ensure active membership in Convention on International Trade in Endangered Species (CITES), Convention for the Conservation of Migratory Species (CMS), RFMOs, and other relevant regional and international agreements;

15. Fully implement and enforce CITES chondrichthyan listings based on solid non-detriment findings, if trade in listed species is allowed;

*Table 7. Continued on next page*

and freshwater) mainly according to depth distribution and, to a lesser degree, position in the water column. The pelagic group includes both neritic (pelagic on the continental shelf) and epipelagic oceanic (pelagic in the upper 200 m of water over open ocean) species. Species habitats were classified based on the findings from the workshops combined with a review of the primary literature, FAO fisheries guides and field guides (*Cavanagh et al., 2003*; *Cavanagh and Gibson, 2007*; *Cavanagh et al., 2008*; *Gibson et al., 2008*; *Camhi et al., 2009*; *Kyne et al., 2012*). Species habitat classifications tended to be similar across families, but for some species the depth distributions often spanned more than one depth category and for these species habitat was assigned according to the predominant location of each species throughout the majority of its life cycle (*Compagno, 1990*). This issue was mainly confined to coastal and continental shelf species that exhibited distributions extending down the continental slopes (e.g., some *Dasyatis*, *Mustelus*, *Rhinobatos*, *Scyliorhinus*, *Squalus*, and *Squatina*). We caution that some of the heterogeneity in depth distribution or unusually large distributions may reflect taxonomic uncertainty and the existence of species complexes (*White and Last, 2012*). We defined the deep sea as beyond the continental and insular shelf edge at depths greater than or equal to 200 m. Coastal and continental shelf includes predominantly demersal species (those spending most time dwelling on or near the seabed), and excluded neritic chondrichthyans. Pelagic species included macrooceanic and tachypelagic ocean-crossing epipelagic sharks with circumglobal distributions as well as sharks suspected of ocean-crossing because they exhibit circumglobal but disjunct distributions, for example Galapagos shark (*Carcharhinus galapagensis*).

Our classification resulted in a total of 33 obligate freshwater and 1,008 marine and euryhaline chondrichthyans of which 482 species were found predominantly in coastal and continental shelf, 39 in pelagic, 479 in deepwater, and eight in meso- and bathypelagic habitats. To evaluate whether the geographic patterns of threat are robust to alternate unique or multiple habitat classifications we considered two alternate classification schemes, one where species were classified into a single habitat and another where species were classified in one or more habitats. The alternate unique classification scheme yielded 42 pelagic (*Camhi et al., 2009*), and 452 deepwater chondrichthyans (*Kyne and Simpfendorfer, 2007*), leaving 517 coastal and continental shelf and 33 obligate freshwater species (totaling 1,044, based on an

*Table 7. Continued*

16. Propose and support the listing of additional threatened chondrichthyan species under CITES and CMS and other relevant wildlife conventions;

17. Collaborate on regional agreements and the CMS migratory shark Memorandum of Understanding (**CMS, 2010**), with a focus on securing concrete conservation actions; and,

18. Strictly enforce chondrichthyan fishing and protection measures and impose meaningful penalties for violations.

older taxonomic scheme). When species were classified in more than one habitat this resulted in 513 species in the coastal and continental shelf, 564 in deepwater, 54 in pelagic, and 13 meso- and bathypelagic habitats. We found the geographic pattern of threat was robust to the choice of habitat classification scheme, and we present only the unique classification (482 coastal and continental shelf, 39 pelagic, 479 deepwater habitat species).

## Major threats

SSG assessors coded each species according to the IUCN Major threat Authority File (http://www.iucnredlist.org/technical-documents/classification-schemes/habitats-classification-scheme-ver3). We coded threats that appear to have an important impact, but did not describe their relative importance for each species.

The term 'bycatch' and its usage in the IUCN Major threat Authority File do not capture the complexity and values of chondrichthyan fisheries. Some chondrichthyans termed 'bycatch' are actually caught as 'incidental or secondary catch' as they are used to a similar extent as the target species or are sometimes highly valued or at least welcome when the target species is absent. 'Unwanted bycatch' refers to cases where the chondrichthyans are not used and fishers would prefer to avoid catching them (Clarke, S personal communication, Sasama Consulting, Shizuoka, Japan). If the levels of unwanted bycatch are severe enough, chondrichthyans can be actively persecuted to avoid negative and costly gear interactions—such as caused the near extirpation of the British Columbian population of Basking shark (*Cetorhinus maximus*) (**Wallace and Gisborne, 2006**).

## Red List Assessment

We assigned a Red List Assessment category for each species based on the information above using the revised 2001 IUCN Red List Categories and Criteria (version 3.1; http://www.iucnredlist.org/technical-documents/categories-and-criteria). We provided a rationale for each assessment justifying the classification along with a description of the relevant criteria used in the designation. Data fields also present the reason for any change in Red List categories from previous assessments (i.e., genuine change in status of species, new information on the species available, incorrect data used in previous assessments, change in taxonomy, or previously incorrect criteria assigned to species); the current population trend (i.e., increasing, decreasing, stable, unknown); date of assessment; names of assessors and evaluators (effectively the peer-reviewers); and any notes relevant to the Red List category. The Red List documentation for each species assessment is supported by references to the primary and secondary literature cited in the text.

## Data entry, review, correction, and consistency checking

Draft regional Red List Assessments and supporting data were collated and peer-reviewed during the workshops and through subsequent correspondence to produce the global assessment for each species. At least one member of the SSG Red List team was present at each of the workshops to facilitate a consistent approach throughout the data collection, review and evaluation process. Once experts had produced draft assessments, SSG staff circulated summaries (comprised of rationales, Red List Categories and Criteria) to the entire SSG network for comment. As the workshops took place over a >10-year period, some species assessments were reviewed and updated at subsequent workshops or by correspondence. Each assessment received a minimum of two independent evaluations as a part of the peer-review process, either during or subsequent to the consensus sessions (a process involving 65 specialists and experts across 23 participating countries) prior to entry into the database and submission to the IUCN Red List Unit.

SSG Red List-trained personnel undertook further checks of all assessments to ensure consistent application of the Red List Categories and Criteria to each species, and the then SSG Co-chair Sarah L Fowler, thoroughly reviewed every assessment produced from 1996 to 2009. Following the data review and evaluation process, all species assessments were entered in the Species Information Service database and checked again by SSG Red List Unit staff. IUCN Red List Program staff made the final

check prior to the acceptance of assessments in the Red List database and publication of assessments and data online (http://www.iucnredlist.org/).

## Subpopulation and regional assessments

We included only global species assessments in this analysis. In many cases, subpopulation and regional assessments were developed for species before a global assessment could be made. For very wide-ranging species, such as the oceanic pelagic sharks, a separate workshop was held to combine these subpopulation or regional assessments (*Table 8*). A numerical value was assigned to each threat category in each region where the species was assessed, and where possible these values were then averaged to calculate a global threat category (*Gärdenfors et al., 2001*). Hence, the Red List categories of some species may differ regionally; for example, porbeagle shark (*Lamna nasus*) is classified as VU globally, but CR in the Northeast Atlantic and Mediterranean Sea. Often population trends were not available across the full distribution of a species. In these cases, the degree to which the qualifying threshold was met was modified according to the degree of certainty with which the trend could be extrapolated across the full geographic range of a species. The calculation of the overall Red List category for globally distributed species is challenging, particularly when a combination of two or more of the following issues occurs: (1) trend data are available only for a part of the geographic range; (2) regional trend data or stock assessments are highly uncertain; (3) the species is data-poor in some other regions; (4) the species is subject to some form of management in other regions; and, (5) the species is moderately productive (*Dulvy et al., 2008*). This situation is typified by the Blue shark (*Prionace glauca*) that faces all of these issues. The best abundance trend data come from the Atlantic Ocean, but the different time series available occasionally yield conflicting results; surveys of some parts of the Atlantic exhibit declines of 53–80% in less than three generations (*Dulvy et al., 2008*; *Gibson et al., 2008*), while a 2008 stock assessment conducted for the International Commission for the Conservation of Atlantic Tuna (ICCAT) indicate, albeit with substantial uncertainty, that the North Atlantic Blue shark population biomass is still larger than that required to generate Maximum Sustainable Yield ($B_{MSY}$) (*Gibson et al., 2008*). The Blue shark is one of the most productive of the oceanic pelagic sharks, maturing at 4–6 years of age with an annual rate of population increase of ~28% per year and an approximate $B_{MSY}$ at ~42% of virgin biomass, $B_0$ (*Cortés, 2008*; *Simpfendorfer et al., 2008*). While the available data may support the regional listing of the Atlantic population of this species in a threatened category, the assessors could not extrapolate this to the global distribution because the species may be subject to lower fishing mortality in other regions. Hence the Blue shark was listed as NT globally. Further details on this issue and additional data requirements to improve the assessment and conservation of such species are considered elsewhere (*Gibson et al., 2008*; *Camhi et al., 2009*).

## Red Listing marine fishes

We assessed most threatened chondrichthyans (81%, n = 148 of 181) using the Red List population reduction over time Criterion A. Only one of the threatened species, the Skate (*Dipturus*) was assessed under the higher decline thresholds of the A1 criterion, where 'population reduction in the past, where the causes are clearly reversible AND understood AND have ceased'. The remaining threatened species were assessed using the IUCN geographic range Criterion B (n = 29) or the small population size and decline Criterion C (n = 4: Borneo shark *Carcharhinus borneensis*, Colclough's shark *Brachaelurus colcloughi*, Northern river shark *Glyphis garricki*, and Speartooth shark *Glyphis glyphis*). The Criterion A decline assessments were based on statistical analyses and critical review of a tapestry of local catch per unit effort trajectories, fisheries landings trajectories (often at lower taxonomic resolution), combined with an understanding of fisheries selectivity and development trajectories.

We assessed most chondrichthyans using the Red List criterion based on population reduction over time (Criterion A). The original thresholds triggering a threatened categorization were Criterion A1: VU 20%; EN 50%; and CR 80% decline over the greater of the past (A1) or future (A2) 10 years or three generations (IUCN Categories and Criteria version 2.3). IUCN raised these thresholds in 2001 to VU, ≥30%; EN, ≥50%; and CR, ≥80% decline over the greater of 10 years or three generations in the past (A2), future (A3) and ongoing (A4), and changed A1 to a reduction over the past 10 yrs or 3 generations of VU ≥50%; EN ≥70%; CR ≥90%, where the causes of reduction are understood AND have ceased AND are reversible. This was in response to concerns that the

**Table 8.** The locations, dates, number of participants and the number of countries represented at each of the SSG Red List workshops, along with unique totals

| Red List workshop | Location | Date | Participants | Countries |
|---|---|---|---|---|
| Australia and Oceania | Queensland, Australia | March 2003 | 26 | 5 |
| South America | Manaus, Brazil | June 2003 | 25 | 8 |
| Sub-equatorial Africa | Durban, South Africa | September 2003 | 28 | 9 |
| Mediterranean | San Marino | October 2003 | 29 | 15 |
| Deep sea sharks | Otago Peninsula, New Zealand | November 2003 | 32 | 11 |
| North and Central America | Florida, USA | June 2004 | 55 | 13 |
| Batoids (skates and rays) | Cape Town, South Africa | September 2004 | 24 | 11 |
| Expert Panel Review | Newbury, UK | March 2005 | 12 | 5 |
| Northeast Atlantic | Peterborough, UK | February 2006 | 25 | 9 |
| West Africa | Dakar, Senegal | June 2006 | 25 | 12 |
| Expert Panel Review | Newbury, UK | July 2006 | 9 | 12 |
| Pelagic sharks | Oxford, UK | February 2007 | 18 | 11 |
| Northwest Pacific/ Southeast Asia | Batangas, Philippines | June/July 2007 | 23 | 13 |
| | | Totals | 227 | 57 |

original thresholds were too low for managed populations that are being deliberately fished down to MSY (typically assumed to be 50% of virgin biomass under Schaeffer logistic population growth) (*Reynolds et al., 2005*). This revision was designed to improve consistency between fisheries limit reference points and IUCN thresholds reducing the likelihood of false alarms—where a sustainably exploited species incorrectly triggers a threat listing (*Dulvy et al., 2005*; *Porszt et al., 2012*). Empirical testing shows that this has worked and demonstrates that a species exploited at fishing mortality rates consistent with achieving MSY ($F_{MSY}$) would lead to decline rates that would be unlikely to be steep enough to trigger a threat categorization under these new thresholds (*Dulvy et al., 2005*).

It is incontrovertible that a species that has declined by 80% over the qualifying time period is at a greater relative risk of extinction than another that declined by 40% (in the same period). Regardless, there may be a wide gap in the population decline trajectory between the point at which overfishing occurs and the point where the absolute risk of extinction becomes a real concern (*Musick, 1999a*). In addition, fisheries scientists have expressed concern that decline criteria designed for assessing the extinction risk of a highly productive species may be inappropriate for species with low productivity and less resilience (*Musick, 1999a*), although this was addressed with the use of generation times to rescale decline rates to make productivity comparable (*Reynolds et al., 2005*; *Mace et al., 2008*). In response to concerns that IUCN decline thresholds are too low and risk false alarms, the American Fisheries Society (AFS) developed alternate decline criteria (*Musick, 1999a*) to classify North American marine fish populations (*Musick et al., 2000*). This approach only categorizes species that have undergone declines of 70–99% over the greater of three generations or 10 years. Nonetheless, most of the species so listed by AFS also appear on the relevant IUCN Specialist Group lists and vice versa, although the risk categories are slightly different. The reason for the concordance is that in most instances the decline had far exceeded 50% over the appropriate timeframe long before it was detected. Consequently, SSG scientists generally agreed in assigning threat categories to species that had undergone large declines, but many were reluctant to assign a VU classification to species that were perceived to be at or near 50% virgin population levels and presumably near $B_{MSY}$. In practice, the latter were usually classified as NT unless other circumstances (highly uncertain data, combined with widespread unregulated fisheries) dictated a higher level of threat according to the precautionary principle.

## Statistical analysis

### Modeling correlates of threat

Vulnerability to population decline or extinction is a function of the combination of the degree to which intrinsic features of a species' behavior, life history and ecology (sensitivity) may reduce the capacity of a species to withstand an extrinsic threat or pressure (exposure). We tested the degree to which intrinsic life histories and extrinsic fishing activity influenced the probability that a chondrichthyan species was threatened. Threat category was modeled as a binomial response variable; with LC species assigned a score of 0, and VU, EN & CR species assigned a 1. We used maximum body length (cm), geographic range size (Extent of Occurrence, $km^2$), and depth range (maximum–minimum depth, m) as indices of intrinsic sensitivity, and minimum depth (m) and mean depth (maximum–minimum depth/2) as a measure of exposure to fishing activity. All variables were standardized to $z$-scores by subtracting the mean and dividing by the standard deviation to minimize collinearity (variance inflation factors were less than 2). Mean depth was not included in model evaluation as it was computed from, and hence, correlated to minimum depth (Spearman's $\rho = 0.52$). We fitted Generalized Linear Mixed-effect Models with binomial error and a logit link to model the probability of a species being threatened, using taxonomic structure as a nested random effect (e.g., order/family/genus) to account for phylogenetic non-independence. The probability of a species $i$ being threatened was assumed to be binomially distributed with a mean $p_i$, such that the linear predictor of $p_i$ was:

$$\log\left(\frac{p_i}{1-p_i}\right) = \beta_0 + \beta_{i,j}X_{i,j} + \beta_{i,k}X_{i,k}, \tag{2}$$

where $\beta_{i,j}$ and $\beta_{i,k}$ are the fitted coefficients for life history or geographic range traits $j$ and $k$, and $X_{i,j}$ and $X_{i,k}$ are the trait values of $j$ and $k$ for species $i$ (**Tables 4 and 9**). The effect of a one standard deviation increase in the coefficient of interest was computed as:

$$1/\left(1 + \exp\left(\beta_0 + \beta_1\right)\right) - 1/\left(1 + \exp\left(\beta_0 + (\beta_1 * 2)\right)\right), \tag{3}$$

following (**Gelman and Hill, 2006**). Models were fitted using the lmer function in the R package lme4 (**Bates et al., 2011**). The amount of variance explained by the fixed effects only and the combined fixed and random effects of the binomial GLMM models was calculated as the marginal **$R^2$**GLMM($m$) and conditional **$R^2$**GLMM($c$), respectively, using the methods described by **Nagakawa and Schlielzeth (2012)**.

### Estimating the proportion of potentially threatened DD species

We predicted the number of Data Deficient species that are potentially threatened based on the maximum body size and geographic distribution traits (**Table 3**; **Supplementary file 1**). Specifically, based on the explanatory models described above, all variables were $\log_{10}$ transformed and we fitted generalized linear models of increasing complexity assuming a binomial error and logit link (**Equation 2**; **Table 3**). Model performance was evaluated using Receiver Operating Characteristics by comparing the predicted probability that the species was threatened $p$(THR) against the true observed status (Least Concern = 0, and threatened [VU, EN & CR] = 1) (**Sing et al., 2005**; **Porszt et al., 2012**). The prediction accuracy was calculated as the Area Under the Curve (AUC) of the relationship between false positive rates and true positive rates, where a false positive is a model prediction of ≥0.5 and true observed status is 0 (or <0.5 and 1) and a true positive is a prediction of ≥0.5 and true observed status is 1 (or <0.5 and 0). True and false positive rates, and accuracy (AUC) were calculated using the R package ROCR (**Sing et al., 2005**). The probability that a DD species was threatened $p$(THR)$_{DD}$ was predicted based on the available life history and distributional traits. DD species with $p$(THR)$_{DD} \geq 0.5$ were classified as threatened and <0.5 as Least Concern. This optimum classification threshold was confirmed by comparing accuracy across the full range of possible thresholds (from 0 to 1). We fitted models using the gls function and calculated pseudo-$R^2$ using the package rms.

With these models we can estimate the number and proportion of species in each category (**Table 1**). We estimated that 68 of 396 DD species are potentially threatened, and hence the remainder (396–68 = 328) is likely to be either Least Concern or Near Threatened. Assuming these species are distributed between these categories according to the observed ratio of NT:LC species of

0.5477 this results in a total of 312 (29.9%) Near Threatened species (132 known + 180 estimated) and 389 (37.4%) Least Concern species (241 known +148 estimated). After apportioning the DD species among threatened (68), NT (312), and LC (389), only 91 (8.7%; 487–396) are likely to be Data Deficient (*Table 1*).

## Spatial analysis

The SSG and the GMSA created ArcGIS distribution maps as polygons describing the geographical range of each chondrichthyan depending on the individual species' point location and depth information. Pelagic species distribution maps were digitized by hand from the original map sources. For spatial analyses, we merged all species maps into a single shapefile. We mapped species using a hexagonal grid composed of individual units (cells) that retain their shape and area (~23,322 km²) throughout the globe. Specifically, we used the geodesic discrete global grid system, defined on an icosahedron and projected to the sphere using the inverse Icosahedral Snyder Equal Area (ISEA) (*Sahr et al., 2003*). A row of cells near longitude 180°E/W was excluded, as these interfered with the spatial analyses (*Hoffmann et al., 2010*). Because of the way the marine species range maps are buffered, the map polygons are likely to extrapolate beyond known distributions, especially for any shallow-water, coastal species, hence not only will range size itself likely be an overestimate, but so will the number of hexagons.

We excluded obligate freshwater species from the final analysis as their distribution maps have yet to be completed. The maps of the numbers of threatened species represent the sum of species that have been globally assessed as threatened, in IUCN Red List categories VU, EN or CR, existing in each ~23,322 km² cell. We caution that this should not be interpreted to mean that species existing within that grid cell are necessarily threatened in this specific location, rather that this location included species that are threatened, on average, throughout their Extent of Occurrence. The number of threatened species was positively related to the species richness of cells ($F_{1, 14,846} = 1.5 \ e^5$, p<0.001, $r^2 = 0.91$). To remove this first-order effect and reveal those cells with greater and lower than expected extinction risk, we calculated the residuals of a linear regression of the number of threatened species on the number of non-DD species (referred to as data sufficient species). Cells with positive residuals were mapped to show areas of greater than expected extinction risk compared to cells with equal or negative residuals. Hexagonal cell information was converted to point features and smoothed across neighboring cells using ordinary kriging using a spherical model in the Spatial Analyst package of ArcView. Such smoothing can occasionally lead to contouring artefacts, such as the yellow wedge west of southern Africa in *Figure 7D*, and we caution against over-interpreting marginal categorization changes.

We identified hotspots of threatened endemic chondrichthyans to guide conservation priorities. To describe the potential cost of losing unique chondrichthyan faunas, we calculated irreplaceability scores for each cell. Irreplaceability scores were calculated for each species as the reciprocal of its area of occupancy measured as the number of cells occupied. For example, for a species with an Extent of Occurrence spanning 100 hexagons, each hexagon in its range would have an irreplaceability 1/100 or 0.01 in each of the 100 hexagons of its Extent of Occurrence. The irreplaceability of each cell was calculated by averaging $\log_{10}$ transformed irreplaceability scores of each species in each cell. Averaging irreplaceability scores controls for varying species richness across cells. We calculated irreplaceability both for all chondrichthyans and for threatened species only. Irreplaceability was also calculated using only endemic threatened species, whereby endemicity was defined as species having an Extent of Occurrence of <50,000, 100,000, 250,000 or 500,000 km². Different definitions of endemicity gave similar patterns of irreplaceability and we present the results of only the largest-scale definition of endemicity. Hence the irreplaceability of threatened species and particularly the threatened endemic chondrichthyans represents those locations or 'hotspots' (*Myers et al., 2000*) at greatest risk of losing the most unique chondrichthyan biodiversity.

## Fisheries catch landings and shark fin exports to Hong Kong

We extracted chondrichthyan landings reported to FAO by 146 countries and territories from a total of 128 countries (as some chondrichthyan fishing nations are overseas territories, unincorporated territories, or British Crown Dependencies) from FishStat (*FAO, 2011*). We categorized landings into 153 groupings, comprised of 128 species-specific categories (e.g., *angular roughshark, piked dogfish, porbeagle, Patagonian skate, plownose chimaera,* small-eyed ray, etc) and 25 broader *nei* (nei = not elsewhere included) groupings (e.g., such as *various sharks nei, threshers sharks nei, ratfishes nei, raja rays nei*). For each country, all chondrichthyan landings in metric tonnes (t) were averaged over the decade 2000–2009. Landings reported as '<0.5' were assigned a value of 0.5 t. Missing data reported as '.' were

**Table 9.** Parameter estimates for General Linear Mixed-effects Models testing the probability that a species is threatened p(THR) given either categorical habitat class or continuous measure of depth distribution and maximum size

**(A) Habitat category**

**p(THR) = maximum length + habitat category, random effect = Order/Family/Genus**

| Fixed effects | Standardized coefficient | Standard error | p-value |
|---|---|---|---|
| Intercept (Coastal and continental shelf) | 0.27 | 0.33 | 0.4 |
| Deepwater | −2.01 | 0.39 | <0.001 |
| Pelagic | −0.46 | 0.94 | 0.62 |
| Maximum length | 2.59 | 0.69 | <0.001 |

marginal $R^2$GLMM($m$) of fixed effects only = 0.40.

conditional $R^2$GLMM($c$) of fixed and random effects = 0.60.

ΔAIC without taxonomic inclusion = −18.7.

ΔAIC for differing threat metrics: binomial THR (CR + EN + VU + NT) = −165.7; categorical = −975.6.

**(B) Minimum depth**

**p(THR) = maximum length + minimum depth, random effect = Order/Family/Genus**

| Fixed effects | Standardized coefficient | Standard error | p-value |
|---|---|---|---|
| Intercept | −0.74 | 0.31 | 0.015 |
| Minimum depth | −2.73 | 0.78 | <0.001 |
| Maximum length | 2.46 | 0.61 | 0.002 |

marginal $R^2$GLMM($m$) of fixed effects only = 0.48.

conditional $R^2$GLMM($c$) of fixed and random effects = 0.64.

ΔAIC without taxonomic inclusion = −12.9.

ΔAIC for differing threat metrics: binomial THR (CR + EN + VU + NT) = −153.4; categorical = −985.8.

**(C) Maximum depth**

**p(THR) = maximum depth + maximum length, random effect = Order/Family/Genus**

| Fixed effects | Standardized coefficient | Standard error | p-value |
|---|---|---|---|
| Intercept | −0.60 | 0.28 | <0.001 |
| Maximum depth | −2.35 | 0.54 | <0.001 |
| Maximum length | 3.03 | 0.63 | <0.001 |

marginal $R^2$GLMM($m$) of fixed effects only = 0.45.

conditional $R^2$GLMM($c$) of fixed and random effects = 0.63.

ΔAIC without taxonomic inclusion = −17.2.

ΔAIC for differing threat metrics: binomial THR (CR + EN + VU + NT) = −156.7; categorical = −981.7.

**(D) Depth range**

**P(THR) = median depth + maximum length, random effect = Order/Family/Genus**

| Fixed effects | Standardized coefficient | Standard error | p-value |
|---|---|---|---|
| Intercept | −0.51 | 0.26 | 0.002 |
| Depth range | −1.82 | 0.50 | <0.001 |
| Maximum length | 3.17 | 0.64 | <0.001 |

marginal $R^2$GLMM($m$) of fixed effects only = 0.42.

conditional $R^2$GLMM($c$) = 0.62.

ΔAIC without taxonomic inclusion = −22.3.

ΔAIC for differing threat metrics: binomial THR (CR + EN + VU + NT) = −158.7; categorical = −982.3.

*Table 9. Continued on next page*

*Table 9. Continued*

**(E) Geographic range (Extent of Occurrence)**

**p(THR) = geographic range + maximum length, random effect = Order/Family/Genus**

| Fixed effects | Standardized coefficient | Standard error | p-value |
|---|---|---|---|
| Intercept | −0.50 | 0.52 | 0.33 |
| Geographic range | 5.22 | 3.7 | 0.12 |
| Maximum length | 2.16 | 0.75 | 0.004 |

marginal $R^2$GLMM($m$) of fixed effects only = 0.65.

conditional $R^2$GLMM(c) = 0.81.

ΔAIC without taxonomic inclusion = −25.8.

ΔAIC for differing threat metrics: binomial THR (CR + EN + VU + NT) = −156.5; categorical = −982.9.

The improvement of model fit by inclusion of phylogenetic random effect was calculated as the difference in AIC (ΔAIC) between the GLMM (with phylogenetic random effect) and a GLM as ΔAIC = AIC(GLMM)-AIC(GLM). $p$(THR) was binomially distributed assuming species that were CR, EN or VU were threatened (1) and LC species were not (0). We present ΔAIC for two other threat classifications, assuming: THR also includes NT species, or THR was a continuous categorical variable ranging from LC = 0 to CR = 5.

assigned a zero. Total annual chondrichthyan landings are underestimated as data are not reported for 1,522 out of a total count of 13,990 entries in the dataset. Therefore, 11% of chondrichthyan landings reported to the FAO over the 10-year period are 'data unavailable, unobtainable'. We mapped FAO chondrichthyan landings as the national percent share of the average total landings from 2000 to 2009.

For the analysis of landings over time we removed the aggregate category 'sharks, rays, skates, etc' and all nine of the FAO chimaera reporting categories. The 'sharks, rays, skates, etc' FAO reported category comprised 15,684,456 tonnes of the reported catch from all countries during 1950–2009, which is a total of 45% of the total reported catch for this time period. However, the proportion of catch in this category has declined from around 50% of global catch to around 35%, presumably due to better reporting of ray catch and as sharks have declined or come under stronger protection (*Figure 1*). The nine chimaera categories make up a small fraction of the global catch, 249,404.5 tonnes from 1950 to 2009, representing 0.72% of the total catch.

Hong Kong has long served as one of the world's largest entry ports for the global shark fin trade. While fins are increasingly being exported to Mainland China where species-specific trade data is more difficult to obtain, each year (from 1996 to 2001) Hong Kong handled around half of all fin imports (*Clarke et al., 2006*). Data on shark fin exports to Hong Kong were requested directly from the Hong Kong Census and Statistics Department (*Hong Kong Special Administrative Region Government, 2011*). We mapped exports to Hong Kong as the proportion of the summed total weight of the four categories of shark fin exported to Hong Kong in 2010: (1) shark fins (with or without skin), with cartilage, dried, whether or not salted but not smoked (trade code: 3055950), (2) shark fins (with or without skin), without cartilage, dried, whether or not salted but not smoked (3055930), (3) shark fins (with or without skin), without cartilage, salted or in brine, but not dried, or smoked (3056940), and (4) shark fins (with or without skin), with cartilage, salted or in brine, but not dried or smoked (3056930). We could not correct the difference in weight due to product type. To identify the threat classification of the chondrichthyan species in the fin trade, we included records of the most numerous species used in the Hong Kong fin trade as well as those species with the most-valued fins (*Clarke et al., 2006, 2007*; *Clarke, 2008*).

## Acknowledgements

We thank the UN Food and Agriculture Organization and John McEachran for providing distribution maps. We thank all SSG staff, interns and volunteers for logistical and technical support: Sarah Ashworth, Gemma Couzens, Kendal Harrison, Adel Heenan, Catherine McCormack, Helen Meredith, Kim O'Connor, Rachel Kay, Charlotte Walters, Lindsay MacFarlane, Lincoln Tasker, Helen Bates and Rachel Walls. We thank Rowan Trebilco, Wendy Palen, Cheri McCarty, and Roger McManus for their comments on the manuscript, and Statzbeerz and Shinichi Nagakawa for statistical advice. Opinions expressed herein are of the authors only and do not imply endorsement by any agency or institution associated with the authors.

Assessing species for the IUCN Red List relies on the willingness of dedicated experts to contribute and pool their collective knowledge, thus allowing the most reliable judgments of a species' status to be made. Without their enthusiastic commitment to species conservation, this work would not be possible. We therefore thank all of the SSG members, invited national, regional and international experts who have attended Regional, Generic and Expert Review SSG Red List workshops, and all experts who have contributed data and their expertise by correspondence. A total of 209 SSG members and invited experts participated in regional and thematic workshops and a total of 302 scientists and experts were involved in the process of assessing and evaluating the species assessments. We express our sincere thanks and gratitude to the following people who have contributed to the GSRLA since 1996. We ask forgiveness for any names that may have been inadvertently omitted or misspelled.

Acuña E, Adams W, Affronte M, Aidar A, Alava M, Ali A, Amorim A, Anderson C, Arauz R, Arfelli C, Baker J, Baker K, Baranes A, Barker A, Barnett L, Barratt P, Barwick M, Bates H, Batson P, Baum J, Bell J, Bennett M, Bertozzi M, Bethea D, Bianchi I, Biscoito M, Bishop S, Bizzarro J, Blackwell R, Blasdale T, Bonfil R, Bradaï MN, Brahim K, Branstetter S, Brash J, Bucal D, Cailliet G, Caldas JP, Camara L, Camhi M, Capadan P, Capuli E, Carlisle A, Carocci F, Casper B, Castillo-Geniz L, Castro A, Charvet P, Chiaramonte G, Chin A, Clark T, Clarke M, Clarke S, Cliff G, Clò S, Coelho R, Conrath C, Cook S, Cooke A, Correia J, Cortés E, Couzens G, Cronin E, Crozier P, Dagit D, Davis C, de Carvalho M, Delgery C, Denham J, Devine J, Dharmadi, Dicken M, Di Giácomo E, Diop M, Dipper F, Domingo A, Doumbouya F, Drioli M, Ducrocq M, Dudley S, Duffy C, Ellis J, Endicott M, Everett B, Fagundes L, Fahmi, Faria V, Fergusson I, Ferretti F, Flaherty A, Flammang B, Freitas M, Furtado M, Gaibor N, Gaudiano J, Gedamke T, Gerber L, Gledhill D, Góes de Araújo ML, Goldman K, Gonzalez M, Gordon I, Graham K, Graham R, Grubbs R, Gruber S, Guallart J, Ha D, Haas D, Haedrich R, Haka F, Hareide N-R, Haywood M, Heenan A, Hemida F, Henderson A, Herndon A, Hicham M, Hilton–Taylor C, Holtzhausen H, Horodysky A, Hozbor N, Hueter R, Human B, Huveneers C, Iglésias S, Irvine S, Ishihara H, Jacobsen I, Jawad L, Jeong C-H, Jiddawi N, Jolón M, Jones A, Jones L, Jorgensen S, Kohin S, Kotas J, Krose M, Kukuev E, Kulka D, Lamilla J, Lamónaca A, Last P, Lea R, Lemine Ould S, Leandro L, Lessa R, Licandeo R, Lisney T, Litvinov F, Luer C, Lyon W, Macias D, MacKenzie K, Mancini P, Mancusi C, Manjaji Matsumoto M, Marks M, Márquez-Farias J, Marshall A, Marshall L, Martínez Ortíz J, Martins P, Massa A, Mazzoleni R, McAuley R, McCord M, McCormack C, McEachran J, Medina E, Megalofonou P, Mejia-Falla P, Meliane I, Mendy A, Menni R, Minto C, Mitchell L, Mogensen C, Monor G, Monzini J, Moore A, Morales M.R.J, Morey G, Morgan A, Mouni A, Moura T, Mycock S, Myers R, Nader M, Nakano H, Nakaya K, Namora R, Navia A, Neer J, Nel R, Nolan C, Norman B, Notarbartolo di Sciara G, Oetinger M, Orlov A, Ormond C, Pasolini P, Paul L, Pegado A, Pek Khiok A.L, Pérez M, Pérez-Jiménez J.C, Pheeha S, Phillips D, Pierce S, Piercy A, Pillans R, Pinho M, Pinto de Almeida M, Pogonoski J, Pollard D, Pompert J, Quaranta K, Quijano S, Rasolonjatovo H, Reardon M, Rey J, Rincón G, Rivera F, Robertson R, Robinson L, J.R, Romero M, Rosa R, Ruíz C, Saine A, Salvador N, Samaniego B, San Martín J, Santana F, Santos Motta F, Sato K, Schaaf-DaSilva J, Schembri T, Seisay M, Semesi S, Serena F, Séret B, Sharp R, Shepherd T, Sherrill-Mix S, Siu S, Smale M, Smith M, Snelson, Jr, F, Soldo A, Soriano-Velásquez S, Sosa-Nishizaki O, Soto J, Stehmann M, Stenberg C, Stewart A, Sulikowski J, Sundström L, Tanaka S, Taniuchi T, Tinti F, Tous P, Trejo T, Treloar M, Trinnie F, Ungaro N, Vacchi M, van der Elst R, Vidthayanon C, Villavicencio-Garayzar C, Vooren C, Walker P, Walsh J, Wang Y, Williams S, Wintner S, Yahya S, Yano K, Zebrowski D & Zorzi G.

## Additional information

### Funding

| Funder | Author |
| --- | --- |
| Conservation International | Sarah L Fowler |
| Packard Foundation | Sarah L Fowler |
| Save Our Seas Foundation | Nicholas K Dulvy |
| UK Department of Environment and Rural Affairs | Sarah L Fowler |
| US State Department | Nicholas K Dulvy, Sarah L Fowler |
| US Department of Commerce | Nicholas K Dulvy |

| Funder | Author |
|---|---|
| Marine Conservation Biology Institute | Sarah L Fowler |
| Pew Marine Fellows Program | Sarah L Fowler |
| Mohamed bin Zayed Species Conservation Foundation | Nicholas K Dulvy |
| Zoological Society of London | Nicholas K Dulvy |
| Canada Research Chairs Program | Nicholas K Dulvy |
| Natural Environment Research Council, Canada | Nicholas K Dulvy |
| Tom Haas and the New Hampshire Charitable Foundation | Sarah L Fowler |
| Oak Foundation | Sarah L Fowler |
| Future of Marine Animal Populations, Census of Marine Life | Sarah L Fowler |
| IUCN Centre for Mediterranean Cooperation | Sarah L Fowler |
| UK Joint Nature Conservation Committee | Sarah L Fowler |
| National Marine Aquarium, Plymouth UK | Sarah L Fowler |
| New England Aquarium Marine Conservation Fund | Sarah L Fowler |
| The Deep, Hull, UK | Sarah L Fowler |
| Blue Planet Aquarium, UK | Sarah L Fowler |
| Chester Zoo, UK | Nicholas K Dulvy, Sarah L Fowler |
| Lenfest Ocean Program | Sarah L Fowler |
| WildCRU, Wildlife Conservation Research Unit, University of Oxford, UK | Sarah L Fowler |
| Institute for Ocean Conservation Science, University of Miami | Sarah L Fowler |
| Flying Sharks | Nicholas K Dulvy |

The funders had no role in study design, data collection and interpretation, or the decision to submit the work for publication.

### Author contributions

NKD, SLF, Conception and design, Acquisition of data, Analysis and interpretation of data, Drafting or revising the article, Contributed unpublished essential data or reagents; JAM, CG, Acquisition of data, Drafting or revising the article; RDC, Conception and design, Acquisition of data, Analysis and interpretation of data, Drafting or revising the article; PMK, LRH, SV, Acquisition of data, Analysis and interpretation of data, Drafting or revising the article; JKC, LNKD, MPF, GHB, SRL, Analysis and interpretation of data, Drafting or revising the article; SVF, JCS, JDS, Analysis and interpretation of data, Drafting or revising the article, Contributed unpublished essential data or reagents; CMP, Acquisition of data, Analysis and interpretation of data; CAS, Conception and design, Analysis and interpretation of data, Drafting or revising the article; KEC, Analysis and interpretation of data, Contributed unpublished essential data or reagents; LJVC, Acquisition of data, Drafting or revising the article, Contributed unpublished essential data or reagents; DAE, MRH, WTW, Drafting or revising the article, Contributed unpublished essential data or reagents

### Additional files

#### Supplementary files

• Supplementary file 1. The Data Deficient chondrichthyan species that are potentially threatened.

• Supplementary file 2. (**A**) IUCN Red List status of chondrichthyans in the fin trade, including (i) families with the most-valued fins, and (ii) the most prevalent species utilized in the Hong Kong fin trade. (**B**) Chondrichthyan species threatened by (i) control measures, and (ii) habitat destruction and degradation, pollution or climate change with the corresponding IUCN threat classification (*Salafsky et al., 2008*). (**C**) Irreplaceable: the 66 threatened endemic sharks and rays ordered in decreasing irreplaceability.

## Major datasets

The following dataset was generated:

| Author(s) | Year | Dataset title | Dataset ID and/or URL | Database, license, and accessibility information |
|---|---|---|---|---|
| The International Union for Conservation of Nature | 2013 | The IUCN Red List of Threatened Species | http://www.iucnredlist.org/ | Publicly available. |

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
