## [Decision Letter]

Thank you for sending your work entitled “Global extinction risk of sharks and rays: drivers, hotspots, and conservation priorities” for consideration at *eLife*. Your article has been favorably evaluated by a Senior editor and 2 peer reviewers, and generated considerable discussion that has unfortunately contributed to a longer response time than usual.

The Senior editor and the two reviewers discussed their comments before we reached this decision, and the Senior editor has assembled the following comments to help you prepare a revised submission.

The manuscript by Dulvy and colleagues provides a global assessment of the status of sharks and rays. This assessment stems from an iterative series of workshops harvesting the judgement and analyses of experts using standardized protocols commonly used by the IUCN and affiliates. Based on this assessment of more than 1000 species, the authors explore the patterning of risk assessments as a function of geography (e.g., depth) and life history (e.g., maximum size) of the species. The resulting data are highly informative, providing a clear picture of the distributions and patterning of threat among this taxon. The value of this work is that it provides the most thorough and comprehensive assessment of global threats to sharks and rays ever produced. It synthesizes the expert judgment of more scientists than any other work and covers more species.

The authors extend beyond their data on assessment of distributions and patterning to explore some of the putative drivers of the patterns, including social, economic, and political factors. While we recognize that we have entered an era where threats to biodiversity and climate force policy makers to demand answers that the scientific community is not ready to produce if it relies on the traditional evidentiary standards of the scientific literature, *eLife* is a life sciences and biomedical journal and we feel uneasy with the amount of opinion expressed in the Results section. Hence, we would ask you to consider the following in preparing a revised submission.

While the distributional assessments are founded in natural scientific rigor, the social scientific assessments are anecdotal and potentially biased. The juxtaposition of scientific rigor and conservation vigor is striking but distracting, particularly when blended in the Results section. The manuscript suffers through this attempt to be all-inclusive and the current submission is disjointed. While there is every effort made to identify data for all species within the taxon, only a subset of the legislation specific to the taxon is reported. By mixing currencies and rigor, the authors are making the manuscript difficult to read, or worse, easy to dismiss.

We either suggest that the more subjective points are vetted from the Results and migrated to the Discussion section, or that a combined Results and Discussion section is created, taking particular care to separate subjective interpretation from strong inferences derived from objective data.

Either way, please give serious consideration to the structure of the manuscript. In its current format, the strength of the analysis is lost within the volume of opinions and prose. Consider, perhaps, a structure of rigorous statistical assessment of the distribution of these species as a function of threat. This alone would be a valuable scientific contribution. The reporting of socioeconomics could be reported elsewhere, perhaps in an accompanying article or within a policy journal.

[Editors' note: further clarifications were requested prior to acceptance, as described below.]

Thank you for resubmitting your work entitled “Extinction risk and conservation of the world's sharks and rays” for further consideration at *eLife*. Your revised article has been favorably evaluated by a Senior editor and two peer reviewers. The manuscript has been improved but there are some remaining issues that need to be addressed before acceptance, as outlined below:

We are pleased to have finally secured reviews of your revision and are ready to accept the manuscript for publication once the two issues detailed below are satisfactorily addressed, namely: 1) the problem about the transparency of the data adjustments used for the different data sets, and 2) the addition of a more explicit, preferably statistically supported consideration of the uncertainty of expert opinions. Thoroughly addressing these two points will neutralize the ease with which skeptical readers can dismiss the main take home message of your paper and additionally will highlight the very distinct possibility that your analysis may be underestimating the severity of the extinction risk.

1) First, there are several places where the presentation applies different adjustments to the data that can be interpreted as modifying data only when it makes the status look worse. These are easily addressed.

2) Second, and a far bigger concern, is the accuracy of estimates from expert opinion. These types of data can have large uncertainties and embedded biases. As the authors note, “Most Chondrichthyan catches are unregulated and often misidentified, unrecorded, aggregated or discarded at sea, resulting in a lack of species-specific landings information.” These same issues raise the critical concern of how much stock we can put into even the opinions of the world's experts. Aside from saying that the experts applied IUCN Red List criteria and used all available data, there is little to help the reader evaluate the level of uncertainty in any of these estimates. As a result, a skeptical reader can easily dismiss much of the findings because of the inherent limits of assessments based largely on expert opinion. There have certainly been evaluations of uncertainty associated with IUCN redlist criteria in other studies, but this issue is largely ignored in the paper except for a discussion related to issues of the category definitions deeply buried in the supplemental materials. This discussion only addresses a few issues affecting uncertainty in these estimates. Although it alludes to some disagreements among experts in terms of assessment status, there is no analysis of the level of uncertainty one should place on these estimates.

There is also the possibility for comparison of the results with other forms of quantitative objective assessments for at least a subset of the data. Without directly addressing this issue of uncertainty somewhere in this paper, the value of this work is compromised and the interpretation of the findings could be misleading. One possible comparison would be with the global fisheries assessments published by Costello et al. in 2012 (Science, 338:517-520). Although they do not have anywhere near as many sharks and cartilaginous species in their analyses (112), they do estimate B/Bmsy for a group of these species using an objective quantitative approach. The median B/Bmsy for previously unassessed species in this analysis was 0.37. If I do a back of the envelope comparison and assume that Bmsy is typically 0.5 * unfished biomass, this would mean that the Costello analysis would have the median shark fishery at about 19% of its unfished biomass. This is actually worse than the values presented in this paper. Here, the median fishery (ignoring the data deficient species) would be at the lower edge of the Near Threatened category since 51% of species with categories are either Least Concerned or Near Threatened. Although this estimate suggests that sharks and rays are better off than the Costello assessment, at least this kind of comparison could be used to make the case that these expert opinion assessments are actually likely to be conservative rather than over-estimating the threatened status of species. Whether is make sense to include such a comparison is up to the authors, but I think the value of this broad extension of assessments to a much larger number of species would gain more credibility if there is a clear consideration of the uncertainty in expert opinions. I suspect that many readers will assume such expert analyses overestimate the nature of the problem. Quite the opposite may be true.

---

## [Author Response]

*We either suggest that the more subjective points are vetted from the Results and migrated to the Discussion section, or that a combined Results and Discussion section is created, taking particular care to separate subjective interpretation from strong inferences derived from objective data*.

*Either way, please give serious consideration to the structure of the manuscript. In its current format, the strength of the analysis is lost within the volume of opinions and prose. Consider, perhaps, a structure of rigorous statistical assessment of the distribution of these species as a function of threat. This alone would be a valuable scientific contribution. The reporting of socioeconomics could be reported elsewhere, perhaps in an accompanying article or within a policy journal*.

Following the guidance of the Senior editor and both reviewers, we have chosen to remove our systematic analysis of social, economic, and political factors, and instead we focus only on the natural science of marine extinction risk. We will seek to publish the social, economic, and political elements of our work elsewhere. We note, however, that all social, economic, and political information was collected in a systematic manner. Hence the anecdotal nature and biased availability of such data reflects the widespread lack of effective management of sharks and rays rather than a lack of scientific rigor. We have compiled the global catch landings trajectory, the trend of increasing ray catch, and the map of catch share and shark fin export into a single figure, which we refer to in the Introduction. This new figure provides important context for our threat analysis. Only in the Discussion have we considered the social, economic, and political context that has resulted in the worldwide elevation of extinction risk faced by chondrichthyans, as reveal here for the first time.

As recommended by the Senior editor and reviewers, we have used this space to elaborate on the statistical modeling of the ecological and geographic correlates of risk. We extended our explanatory statistical modeling to develop predictive models (Table 3). We use these predictive models to classify and identify those Data Deficient sharks and rays are likely to be threatened based on their body size and degree of accessibility to fisheries (Supplementary file 1). These models explain between 58 and 80% of the variance in species extinction risk (Tables 3 and 4).

[Editors' note: further clarifications were requested prior to acceptance, as described below.]

*1) First, as discussed in minor comments, there are several places where the presentation applies different adjustments to the data that can be interpreted as modifying data only when it makes the status look worse. These are easily addressed*.

We have addressed this issue in response to the substantive comment below.

*2) Second, and a far bigger concern, is the accuracy of estimates from expert opinion. These types of data can have large uncertainties and embedded biases. As the authors note, “Most Chondrichthyan catches are unregulated and often misidentified, unrecorded, aggregated or discarded at sea, resulting in a lack of species-specific landings information.” These same issues raise the critical concern of how much stock we can put into even the opinions of the world's experts. Aside from saying that the experts applied IUCN Red List criteria and used all available data, there is little to help the reader evaluate the level of uncertainty in any of these estimates. As a result, a skeptical reader can easily dismiss much of the findings because of the inherent limits of assessments based largely on expert opinion. There have certainly been evaluations of uncertainty associated with IUCN redlist criteria in other studies, but this issue is largely ignored in the paper except for a discussion related to issues of the category definitions deeply buried in the supplemental materials. This discussion only addresses a few issues affecting uncertainty in these estimates. Although it alludes to some disagreements among experts in terms of assessment status, there is no analysis of the level of uncertainty one should place on these estimates*.

*There is also the possibility for comparison of the results with other forms of quantitative objective assessments for at least a subset of the data. Without directly addressing this issue of uncertainty somewhere in this paper, the value of this work is compromised and the interpretation of the findings could be misleading. One possible comparison would be with the global fisheries assessments published by Costello et al. in 2012 (Science, 338:517-520). Although they do not have anywhere near as many sharks and cartilaginous species in their analyses (112), they do estimate B/Bmsy for a group of these species using an objective quantitative approach. The median B/Bmsy for previously unassessed species in this analysis was 0.37. If I do a back of the envelope comparison and assume that Bmsy is typically 0.5 * unfished biomass, this would mean that the Costello analysis would have the median shark fishery at about 19% of its unfished biomass. This is actually worse than the values presented in this paper. Here, the median fishery (ignoring the data deficient species) would be at the lower edge of the Near Threatened category since 51% of species with categories are either Least Concerned or Near Threatened. Although this estimate suggests that sharks and rays are better off than the Costello assessment, at least this kind of comparison could be used to make the case that these expert opinion assessments are actually likely to be conservative rather than over-estimating the threatened status of species. Whether is make sense to include such a comparison is up to the authors, but I think the value of this broad extension of assessments to a much larger number of species would gain more credibility if there is a clear consideration of the uncertainty in expert opinions. I suspect that many readers will assume such expert analyses overestimate the nature of the problem. Quite the opposite may be true*.

We agree with the referee and appreciate the efforts to bring our attention to new literature. While we were aware of the Costello et al. paper, we were not aware that it contained this vital and alarming estimate of the sustainability of 112 shark and ray fisheries. These data do indeed suggest our analyses maybe conservative, we have also found subsequent trend data from the Mediterranean Sea that also suggest that our Red List Assessments are conservative and err on the side of underestimating risk. We have added two new paragraphs comparing our findings to both the [30] paper and the [44] paper.